# Epigenomic landscape of human colorectal cancer unveils an aberrant core of pan-cancer enhancers orchestrated by YAP/TAZ

Giulia Della Chiara[1,2,3,4,24], Federica Gervasoni [1,2,3,4,24], Michaela Fakiola [1,2,24], Chiara Godano[2,3,5,24], Claudia D'Oria[1,2,3,4], Luca Azzolin[6], Raoul Jean Pierre Bonnal [1,2], Giulia Moreni[2,21], Lorenzo Drufuca[1,2], Grazisa Rossetti[1,2], Valeria Ranzani[2], Ramona Bason[1,2,4], Marco De Simone[2,22], Francesco Panariello [4,23], Ivan Ferrari[4], Tanya Fabbris[2], Francesca Zanconato[6], Mattia Forcato [7], Oriana Romano [7], Jimmy Caroli [7], Paola Gruarin[2], Maria Lucia Sarnicola[2], Michelangelo Cordenonsi [6], Alberto Bardelli[8,9], Nicola Zucchini[10], Andrea Pisani Ceretti[11], Nicolò Maria Mariani [11], Andrea Cassingena[12], Andrea Sartore-Bianchi [12,13], Giuseppe Testa [3,13,14], Luca Gianotti[15], Enrico Opocher[11,16], Federica Pisati[17], Claudio Tripodo[18,19], Giuseppe Macino[20], Salvatore Siena [12,13], Silvio Bicciato [7], Stefano Piccolo [1,6✉] & Massimiliano Pagani [1,2,3,4✉]

Cancer is characterized by pervasive epigenetic alterations with enhancer dysfunction orchestrating the aberrant cancer transcriptional programs and transcriptional dependencies. Here, we epigenetically characterize human colorectal cancer (CRC) using de novo chromatin state discovery on a library of different patient-derived organoids. By exploring this resource, we unveil a tumor-specific deregulated enhancerome that is cancer cell-intrinsic and independent of interpatient heterogeneity. We show that the transcriptional coactivators YAP/TAZ act as key regulators of the conserved CRC gained enhancers. The same YAP/TAZ-bound enhancers display active chromatin profiles across diverse human tumors, highlighting a pan-cancer epigenetic rewiring which at single-cell level distinguishes malignant from normal cell populations. YAP/TAZ inhibition in established tumor organoids causes extensive cell death unveiling their essential role in tumor maintenance. This work indicates a common layer of YAP/TAZ-fueled enhancer reprogramming that is key for the cancer cell state and can be exploited for the development of improved therapeutic avenues.

A list of author affiliations appears at the end of the paper.

Colorectal cancer (CRC) features amongst the three most widely spread malignancies worldwide[1], characterized by diverse clinical phenotypes and responses to current treatments. Indeed, one of the most prominent CRC features is its considerable interpatient heterogeneity. Recent years have seen numerous efforts to classify genetically and phenotypically diverse CRC tumors into distinct molecular subtypes based on gene expression profiling[2,3]. These studies highlight the challenges in identifying a consensus molecular classification system and the urgent need to re-assess interpatient variability through the prism of a shared regulatory architecture.

Epigenetic deregulation has emerged as a paradigm of cancer biology that underlies the hallmarks of cancer cells[4]. Global changes in DNA methylation, chromatin states, and *cis*-regulatory elements, as well as genetic aberrations in chromatin proteins characterize more than 50% of human cancers[5]. Enhancers, identified with precision via global mapping of histone modifications, contribute to cell reprogramming towards tumor growth and metastasis[6,7], rendering enhancer dysfunction a promising biomarker of diagnosis and a potential therapeutic target. Indeed the deregulated enhancerome has attracted attention due to its role in the establishment and maintenance of transcriptional addiction[8], a state of tumor cell dependence on transcription factors and chromatin regulators for sustained uncontrolled proliferation.

In this line of investigation, many questions remain underexplored: what is the enhancer repertoire activated in cancer cells and is it tumor-specific or, perhaps, is there a common layer of epigenetic deregulation across diverse human malignancies? Systematic studies of chromatin modifications in cancer[9] and seminal reports on the molecular mechanisms at the roots of transcriptional addiction[10] have started to emerge. Nevertheless, defining the chromatin states that characterize a human pathology remains a challenge, partly due to the limited amount of cells available from primary tumors. Another critical issue in investigating the epigenetic profiles of individual tumors is the use of appropriate research platforms that capture the cancer-cell intrinsic profiles[11]. Three-dimensional, self-organizing organoid cultures represent such powerful ex vivo models that, contrary to cell lines, recapitulate the overall architecture and functional features of the tissues from which they originate[12]. In this context, patient-derived organoids (PDOs) of CRC capture the histological and molecular heterogeneity of pure cancer cells[13] offering an amenable tool to decipher the underlying regulatory networks of primary tumors.

Here, we seek to identify the common epigenetic blueprints that may affect the cellular events mediating cancer and provide leads on the master regulators that orchestrate the transcriptional deregulation, upon which cancer cells depend. Leveraging the organoid technology, we generate a balanced library of heterogeneous PDOs from CRC patients and decipher their histone modification landscapes through de novo chromatin state reconstruction. Our study provides a detailed resource catalogue of human CRC-specific epigenetic features, enabling the identification of a conserved active enhancerome independent of interpatient tumor heterogeneity. We further highlight the transcriptional coactivators YAP/TAZ as key drivers of the gained enhancers. Of critical relevance to cancers beyond CRC, we show that a core of the YAP/TAZ-fueled deregulated enhancers is consistently active in diverse tumor types crossing the cellular and molecular divides of tissue of origin, genetic aberrations, and microenvironmental stimuli.

epigenetic landscape of human CRC. As an overview of our approach (Fig. 1a), we collected patients' samples and used them to generate organoids, performed histopathological and molecular characterization of organoids validating them as surrogates of the primary tumors from which they derive, and established their genome-wide epigenetic landscape (Supplementary Table 1).

In the first step, we obtained a pool of PDOs recapitulating the molecular heterogeneity of human CRC. For this, we performed RNA-sequencing (RNA-seq) on the primary tumors (from surgical resection) and screened them for markers of microsatellite instability (MSI) and three recently published gene expression classifiers[2,3,14], contextually generating PDOs from the same donors. Upon primary tumor analysis, we selected a collection of 10 PDO lines representing a balanced library that recapitulates the molecular diversity of the cancer-cell intrinsic features of human primary CRCs (Fig. 1b).

Next, we validated that our PDOs preserve the histopathological and molecular features of primary tumors. We first tested whether PDOs retained the typical morphological characteristics and the deregulated architecture of crypt/villus-like structures of human CRC. By using 3D immunofluorescence whole-mount analysis, PDOs displayed disorganized epithelial polarity (EpCAM and F-actin staining respectively, Fig. 1c, first row), random distribution of cell proliferation (Ki67, Fig. 1c, first row), displaced localization of enterocytes (FABP1, Fig. 1c, second row) and presence of cytokeratin 20 positive cells (KRT20, Fig. 1c, third and fourth rows), recapitulating the common dysplastic features of human CRC. Interestingly, goblet-specific mucin 2 (MUC2) was absent in most PDOs but massively produced in the organoid derived from the mucinous adenocarcinoma of patient 13 (Fig. 1c, third and fourth rows), consistent with the histopathological features of the clinical specimen (Supplementary Table 1). Upon passaging and long-term culturing, PDOs retain stable morphology and proliferation rate, and remain transcriptionally stable (see below and Supplementary Fig. 1a).

Next, we used RNA-seq to compare PDOs with the original CRCs and normal counterpart from the same patient (Supplementary Data 1). As shown in Fig. 1d, principal component analysis (PCA) revealed that the transcriptional specificities of each individual tumor, captured by the second principal component (PC2), are preserved in its corresponding PDOs ("Methods" and Supplementary Data 2). Globally, 84% of expressed genes were concordant between PDOs and tumors (Fig. 1e, Venn diagram; Supplementary Fig. 1b, c) with the expression levels across genes being well correlated between PDOs and primary tumors (Fig. 1e, correlation plot, and Supplementary Fig. 1d). We then sought to determine which genes are differentially regulated (induced and repressed) in primary tumor samples when compared to normal mucosa to identify a CRC signature and validate its expression in PDOs. Hierarchical clustering analysis revealed that the CRC signature groups PDOs together with primary tumors and separates them from normal colon mucosa (Fig. 1f). Gene set enrichment analysis (GSEA), using gene signatures of colon carcinoma patients[15], further confirmed that PDOs, when compared to normal tissues, were enriched in tumor-related genes and depleted in normal mucosa genes (Fig. 1g). We conclude from these experiments that PDOs closely recapitulate the cellular and molecular features of their original CRC. Thus, they offer an abundant physiological resource of pure cancer cells from primary patient samples that allow us to focus on the tumor-derived epigenetic profiles uncoupled by those of non-tumoral cells.

## Results

**PDO library as a model of human CRC heterogeneity.** With this background in mind, here we set to characterize the

**De novo chromatin state discovery in human CRC.** Data presented above establish that our PDO library of different CRC

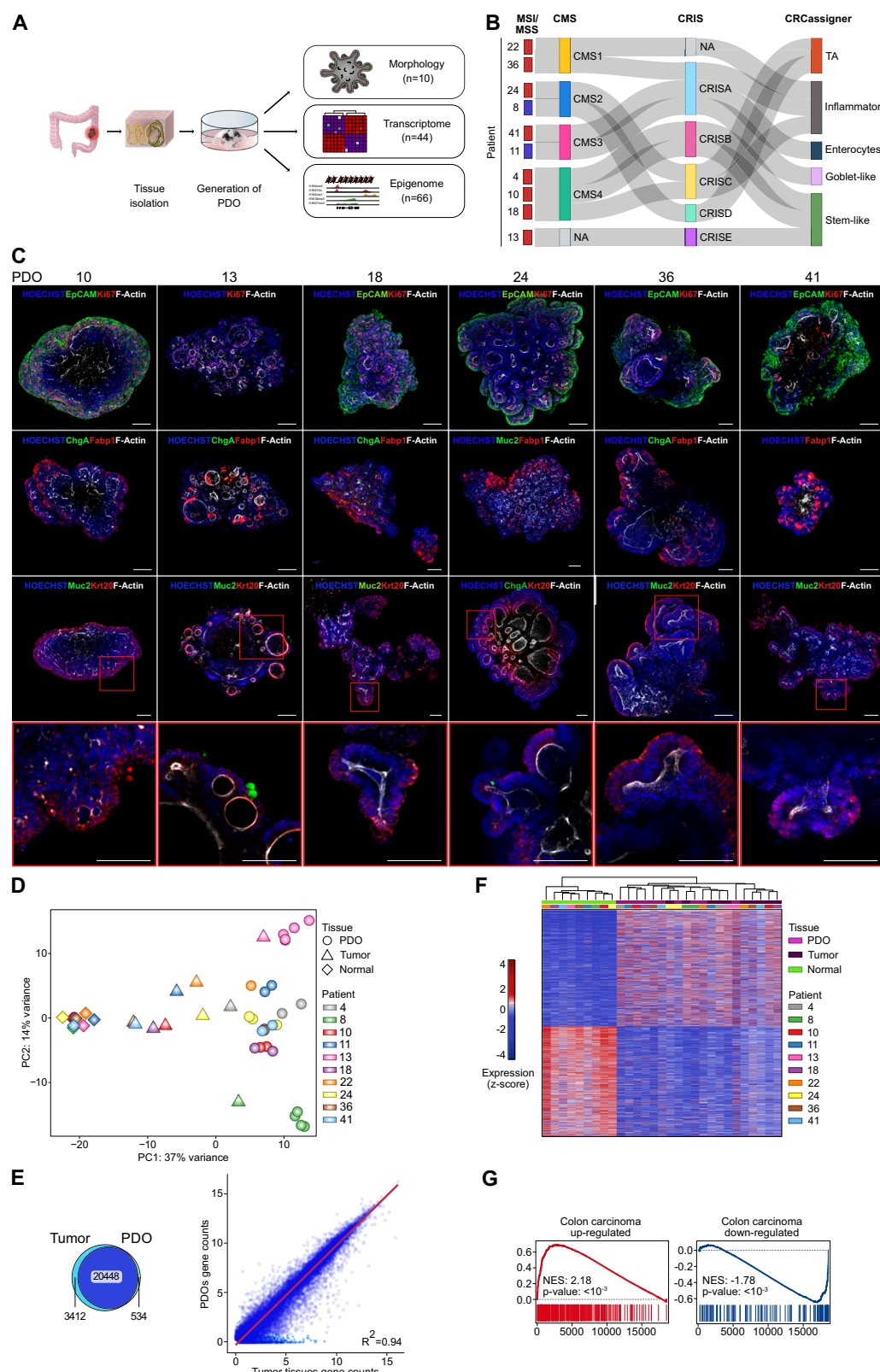

subtypes is suitable for deciphering the epigenetic blueprints of the CRC-cell intrinsic phenotype. Towards this aim, we first performed a multi-factorial integrative analysis of genome-wide chromatin immunoprecipitation sequencing (ChIP-seq) for a core set of five histone modifications (H3K4me3, H3K27ac, H3K4me1, H3K36me3, and H3K27me3) on all PDOs (Fig. 2a and Supplementary Data 1). Unsupervised hierarchical clustering and

principal component analyses (Fig. 2b and Supplementary Fig. 2a) revealed a clear division between histone marks that represent the major types of chromatin regulation: the repressive marker H3K27me3, the elongation marker H3K36me3, and the block of histone marks defining active regulatory regions (H3K4me3, H3K27ac, and H3K4me1), with ChIP-seq data of PDOs for each of these histone marks being clustered together.

**Fig. 1 Establishment of a CRC PDO library that recapitulates the heterogeneity of the primary tumors. a** Schematic depicting the generation of a patient-derived organoid (PDO) library from primary CRC tissues and its subsequent morphological, transcriptomic, and epigenomic profiling. **b** Genetic and molecular classification of primary tumors revealed the heterogeneity of the organoid library. CRC patients are classified based on the consensus molecular subtypes[2], the CRC intrinsic classifier[3], and the CRCassigner[14]. MSI Microsatellite instable (blue), MSS Microsatellite stable (red), CMS Consensus molecular subtypes, CRIS CRC intrinsic classifier. **c** Representative confocal images of 3D immunofluorescence whole-mount analysis on human CRC PDOs. Different markers of colon cell types are shown: polarity and structure (F-Actin, first row), the epithelium (EpCAM, first row), proliferation (Ki67, first row), absorptive cells (FABP1, second row), enteroendocrine cells (CHGA, second and third row), goblet cells (MUC2, third row), and top epithelial crypt cells (KRT20, third row). The fourth row provides an enlargement of the boxed area in the third row. Scale bars, 100 μm. **d** PCA on normalised gene counts from RNA-seq data distinguished normal colon mucosa, primary tumor, and PDOs. **e**, Venn diagram showing the number of concordant expressed genes between tumors and PDOs. Mean log2 normalized gene counts between primary tumors and PDOs were well correlated (Pearson correlation). See Supplementary Fig. 1b–d. **f** Hierarchical clustering analysis using differentially expressed genes (DEG, two-sided adjusted P-value ≤ 0.01 by Wald test with Benjamini–Hochberg false discovery rate correction) between tumor and normal colon tissues clustered PDOs together with parental tumors. Tissue populations and patients are represented by color-coded bars above the heatmap. **g** PDOs are enriched in gene signatures of CRC clinical specimen[15]. GSEA on the ranked list of genes from the comparison between PDOs and normal colon tissue with the normalized enrichment score (NES) and nominal P-value using 1000 permutations reported.

Notably, the same applied to CRC tumors displaying MSI, typified by a bewildering amount of genetic aberrations (PDO8 and PDO11).

To capture the epigenetic layer of CRC complexity in a systematic manner we exploited machine learning approaches to perform de novo chromatin state characterization on the complete ChIP-seq data for the PDOs (Supplementary Data 3). Using ChromHMM[16], we explored the combinatorial patterns of the five histone marks in an 8-state model and predicted specific genomic features with high resolution across samples (Fig. 2c and Supplementary Fig. 2b–d). We defined two promoter states ("Active TSS" and "Flanking Active TSS"), one weak and two active enhancer states ("Weak Enhancers", "Flanking Active Enhancers", and "Active Enhancers"), as well as an elongation, a repressed, and a quiescent state. We then compared the ChromHMM data with chromatin accessibility for colon adenocarcinoma (COAD) using ATAC-seq (Assay for Transposase Accessible Chromatin with high-throughput sequencing) datasets obtained from The Cancer Genome Atlas (TCGA)[17]. The chromatin states identified in PDOs remarkably concur with chromatin accessibility, with active states displaying the highest and more inactive regions the lowest ATAC-seq signals, respectively (Fig. 2d). This provides further support that PDOs preserve the regulatory networks of primary tumors and thus represent a faithful resource to investigate the epigenetic landscape of CRC. Importantly, the ChromHMM-defined chromatin states of CRCs constitute a precise atlas of genome-wide regulatory elements that enables the functional interpretation of ATAC-seq-defined open chromatin regions.

Human CRCs are characterized by extensive cellular and molecular heterogeneity. Yet, they also share a considerable fraction of their transcriptional (Fig. 1f) and epigenetic profiles (Fig. 2b and Supplementary Fig. 2a). Epigenetic conservation is shown for *FABP1*, a marker of enterocytic differentiation expressed by all PDOs (Fig. 1c, second row); all PDOs display an active chromatin state at this locus as visualized by the peaks associated with active histone marks (H3K4me3, H3K27ac, and H3K4me1) at the promoter and flanking regions (Fig. 2e). Occurrences of epigenetic intertumor heterogeneity were also present in our analyses. Regulatory variability across PDOs was observed in the gene encoding for laminin subunit α-5 (LAMA5) (Fig. 2f and Supplementary Fig. 3a), a marker of cell adhesion and migration reported to be involved in metastasis[18]. Active transcription in PDO11 is indicated by a ChromHMM profile that associates with active states around the transcription start site (TSS) and with an elongation state along the gene body. On the contrary, *LAMA5* is silenced in PDO18 evident by the loss of H3K36me3 and the accumulation of the H3K27me3 repressive mark at the promoter and throughout the gene body. The tumor-

specific epigenetic features displayed for *LAMA5* in PDOs 11 and 18 are also mirrored at the protein level by 3D immunofluorescence analysis (Supplementary Fig. 2e), with active and repressed chromatin profiles associated to protein presence or loss, respectively. PDO11 also showed an abundant expression of the goblet cell-specific marker MUC2 and an active epigenetic state across the gene (Supplementary Fig. 2e), suggesting a mucinous phenotype that is consistent with the MSI status of this tumor[19].

This de novo chromatin state characterization provided a rich reference resource (available at https://bioinformatics.ifom.eu/hepic/) of regulatory elements (promoters and enhancers), elongating and repressed genomic regions defined by multiple epigenetic features, enabling a comprehensive interrogation of the colon cancer epigenetic landscape.

**Enhancerome definition of human CRC.** We next sought to gain further insights into the colon cancer enhancerome taking advantage of this systematic reconstruction of the CRC epigenome. To this end, we used the ChromHMM-defined enhancer states, primarily enriched in H3K27ac and H3K4me1 (Fig. 2c), to identify active distal enhancer regions in PDOs and normal colon tissue. We found a total of 33,131 enhancers observed in at least two PDOs and/or normal tissues and located >5 Kb from TSS (Supplementary Data 4), covering 3% of the human genome. Unsupervised clustering of the H3K27ac signals for the ~33 K enhancers supported the discrete separation of the PDO enhancerome from that of normal colon tissues (Fig. 3a). Genome-wide analysis of H3K27ac signals in the ChromHMM-defined enhancers, identified 7828 differentially enriched enhancers (gained or lost) with >4-fold change (adjusted P-value < 0.01) in H3K27ac between PDOs and normal tissues (Fig. 3b). Of these, 2419 enhancers (see Supplementary Fig. 4a for the genomic distribution) were specifically activated (gained) in PDOs and 5409 in normal tissues.

To identify common epigenetic blueprints across the organoid library, we looked at the concordance of tumor-enriched enhancers in the PDOs. Among the gained enhancers, 20% were concordant in 8 to 10 PDOs (n = 486 enhancers), while half of them were shared by at least 5 patients (Fig. 3c and Supplementary Fig. 4b, c), independently of their original molecular and histological features (Fig. 1b and Supplementary Table 1). An example of gained enhancers is shown for *PHLDA1*, a gene upregulated in colon cancer and involved in tumor cell proliferation and migration[20]. *PHLDA1* displayed a conserved epigenetic profile with peaks of H3K27 acetylation acquired in enhancers across all PDOs compared to normal colon tissue (Fig. 3d, vertical boxes). Conversely, the reduced expression of

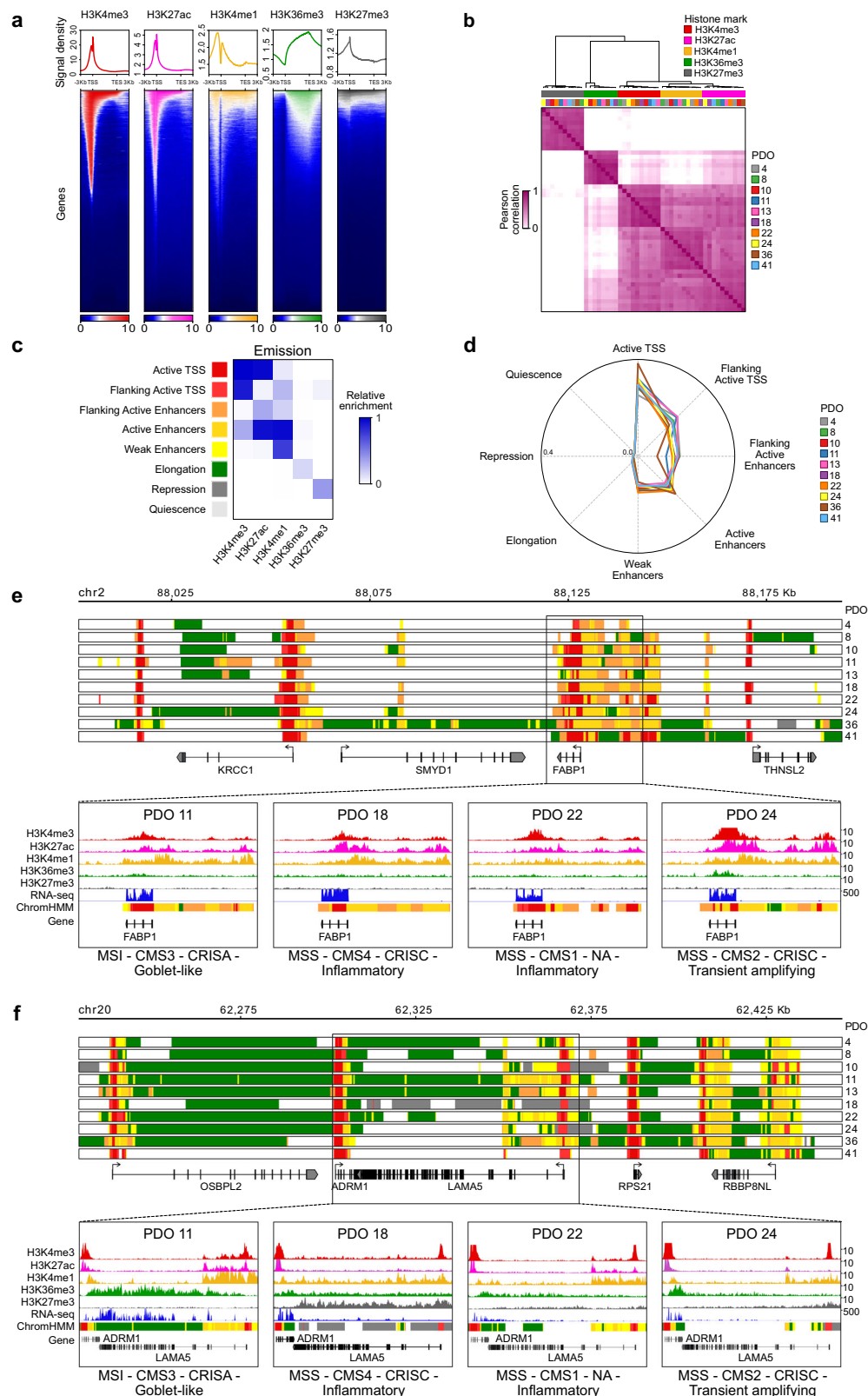

*MUC4* during CRC progression is mirrored by the loss of enhancer activity in all PDOs, as underlined by the absence of active enhancers upstream of *MUC4* (Fig. 3e). Overall, focusing on ChromHMM-defined active enhancers, we characterized the CRC enhancerome and identified the gained enhancers that are shared across our heterogeneous PDOs.

**YAP/TAZ are key regulators of the conserved CRC enhancerome.** Deciphering transcription factor occupancy at enhancer regions is essential for understanding the transcriptional regulatory networks. Motif discovery identified the TEA domain family members (TEAD) as enriched within the accessible regions of the conserved (8–10 patients) gained enhancers

**Fig. 2 Characterization of the human CRC epigenome using ChromHMM analysis. a** Histone modifications localization in relation to the gene body as well as regions surrounding ± 3 Kb of the transcription start (TSS) and end (TES) sites. Representative density plots of average intensity (top) and corresponding heatmaps (bottom) display the relative distribution of H3K4me3 (red), H3K27ac (pink), H3K4me1 (yellow), H3K36me3 (green), and H3K27me3 (gray) signals for all the genes present in the GENCODEv25 annotation. **b** Pearson correlation heatmap of ChIP-seq data for the complete set of five histone modifications across all patient-derived organoids (PDOs). See Supplementary Fig. 2a. **c** Combinatorial pattern of histone marks in an 8-state model using ChromHMM. The heatmap (Emission plot) displays the frequency of the histone modifications found in each state (Supplementary Fig. 2b). **d** The probability of each ChromHMM-defined chromatin state overlapping ATAC-seq regions for TCGA colon adenocarcinoma samples is shown across PDOs using a spider plot. **e–f** Representative tracks of ChromHMM states for the (**e**) FABP1 and (**f**) LAMA5 (Supplementary Fig. 3a) genomic loci in all PDOs. The tracks denote regions identified as promoter (red), active enhancer (orange), weak enhancer (yellow), elongation (green), repressed (gray) or quiescent (white) states (Fig. 2c). The expanded regions show H3K4me3, H3K27ac, H3K4me1, H3K36me3, and H3K27me3 profiles, along with RNA-seq signal and ChromHMM states for PDOs of different molecular subtypes as indicated.

(Supplementary Data 5), suggesting a role for these factors and their transcriptional coactivators YAP/TAZ, as putative regulators of the conserved CRC enhancerome. This is further supported by the enrichment of motifs for AP-1 factors (i.e., Jun and Fos family members) in the same enhancers. Indeed, AP-1 has been recently established as an intimate partner of YAP/TAZ/TEAD, co-occupying disproportionally and pervasively cis-regulatory regions also bound by YAP/TAZ and TEAD[21].

Taking a different approach in identifying the main regulators of the CRC enhancerome, we sought to determine which of the genes annotated to gained enhancers were upregulated in the majority of PDOs compared to normal tissues (P-adjusted < 0.05, based on differential expression analysis of RNA-seq data). We confirmed that the same genes were also upregulated in primary tumors (Supplementary Fig. 4d). Active enhancers were assigned to their putative target genes based on chromosome conformation capture (capture Hi-C) data on human colon cancer[22] or using the nearest protein-coding gene overlapping a ChromHMM-defined active promoter state (Supplementary Data 6). Notably, functional enrichment analysis using the tumor-upregulated enhancerome genes identified the Hippo signaling pathway as the most significantly enriched pathway (Fig. 3f, g), confirming the involvement of the Hippo pathway downstream effectors YAP/TAZ in the activation of the CRC-specific active enhancers.

YAP/TAZ are found to be stably activated in CRC and other types of cancer[23]. To validate the role of YAP/TAZ in the regulation of the CRC enhancerome, we first examined their expression. Both YAP and TAZ were transcriptionally upregulated in PDOs and primary tumors compared to their normal counterpart (Fig. 4a), with TAZ showing the largest difference in gene expression levels compared to the normal tissue (log2FC > 7). Using RNA-seq data of COAD samples from the TCGA dataset, we found that YAP/TAZ expression is also significantly higher (P < 0.0001, Wilcoxon rank sum test) in tumor compared to normal tissue samples (Supplementary Fig. 5a). Immunohistochemistry analysis confirmed the presence of the active form of the YAP and TAZ proteins in the nuclei of PDOs, as well as in primary tumors compared to matched normal tissues (Fig. 4b), all in all confirming the hyperactivation of these factors in CRC.

Using ChIP-seq, we generated a genomic map of TAZ recruitment to the chromatin of PDOs. We identified 14,878 TAZ peaks distributed across ChromHMM-defined active enhancers or promoters (Fig. 4c and Supplementary Data 7). Representative examples of TAZ enrichment across the active promoters of Hippo signaling canonical target genes are shown in Fig. 4d and Supplementary Fig. 5b. The genomic distribution of TAZ on target genes was also mirrored by the YAP ChIP-seq profile (Supplementary Fig. 5b). Since YAP/TAZ do not bind directly to DNA[24], we searched for transcription factors binding motifs encompassing the summit of YAP/TAZ peaks and we found the TEAD family binding motif amongst the most enriched

(Fig. 4e), confirming the cooperative interaction of TEAD transcription factors with YAP/TAZ in human CRC[21].

To better characterize the genomic occupancy of YAP/TAZ, we integrated ChIP-seq data for TAZ with all the CRC chromatin states defined by ChromHMM. TAZ was enriched in active regulatory states with more than 95% of TAZ peaks being present in either promoters or enhancers (Fig. 4f), confirming the active role of this transcriptional activator in CRC. Focusing on the gained CRC-enhancerome, we assessed the overlap of TAZ peaks with the total number of gained enhancers as well as those conserved in at least 50 or 80% of PDOs. Notably, the percentage of enhancer regions bound by TAZ increased with the level of conservation between PDOs (Fig. 4g) up to 40% in the highly conserved enhancers (n = 195, in 8 to 10 PDO lines, Supplementary Data 8) indicating a role for TAZ in regulating a sizeable fraction of the shared CRC enhancerome. In support, the TAZ signal intensity at the conserved enhancers is significantly higher (P = 0.028, Mann–Whitney U test) compared to that at non-conserved gained enhancers (shared by <5 PDOs) (Supplementary Fig. 5c).

Interestingly, TAZ itself is regulated by a TAZ-bound intronic enhancer (Fig. 4h, boxed area) that is shared by all PDOs. The enrichment of TAZ at its own promoter and intronic enhancer suggests a previously unnoticed feedback loop driving its transcriptional activation. Another gene regulated by a TAZ-bound enhancer is epiregulin (EREG), a ligand of the epidermal growth factor (EGF) receptor, that is highly expressed in various tumors[25]. EREG is a known target of the Hippo signaling pathway, linked to intestinal stem cell maintenance and YAP function[26]. The high expression of EREG in PDOs was regulated by two separate enhancers located more than 200Kb downstream of the gene and occupied by TAZ and the active histone marks H3K4me1, H3K27ac, and H3K4me3 (Supplementary Fig. 5d). Another gene of the TAZ-driven CRC enhancerome is the Forkhead box Q1 (FOXQ1) member of the forkhead transcription factor family, which is involved in CRC tumorigenicity and growth[27] with no previously reported role as a YAP/TAZ downstream target gene. The high expression of FOXQ1 in PDOs was controlled by a highly active enhancer bound by both TAZ and YAP (Supplementary Fig. 5e). In situ hybridization in primary healthy and tumor colon tissues confirmed that FOXQ1 gene expression is restricted to the CRC sections that express YAP in the nucleus (Supplementary Fig. 5f). Taken together, these results revealed that the conserved cancer-cell intrinsic CRC-enhancerome, shared by all PDOs and thus independent of patient-to-patient tumor molecular diversity, is largely under the regulatory control of YAP/TAZ, suggesting that these transcriptional activators serve as central players in the epigenetic deregulation of CRC. The role of YAP/TAZ as regulators of the conserved enhancerome of human CRC nicely parallels functional evidence in vivo, in mouse models, and in tumor organoids bearing YAP/

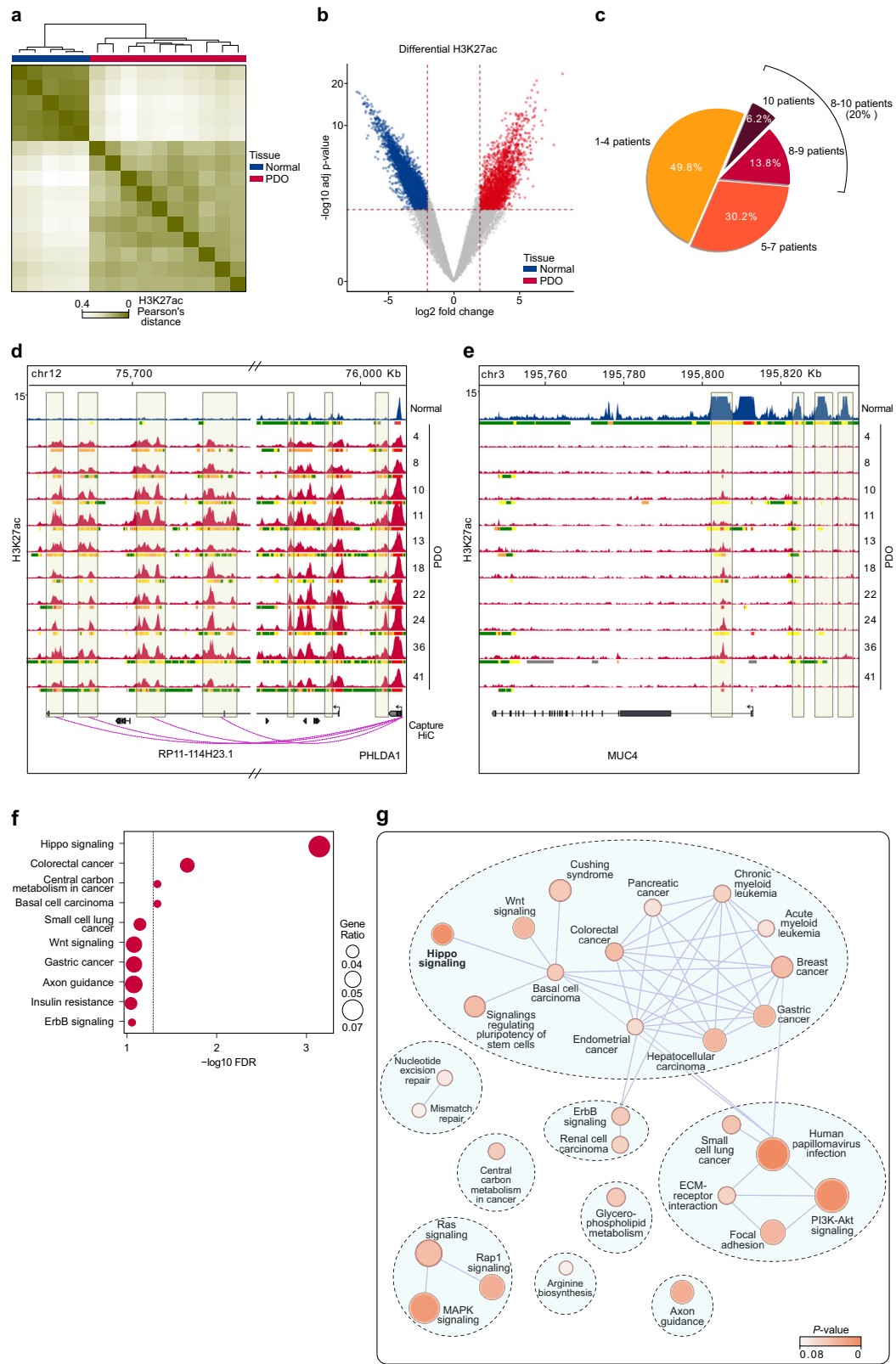

TAZ knockout alleles, all in all indicating the centrality of YAP/TAZ activation for CRC initiation and progression[26,28,29].

**YAP/TAZ are required for maintenance of tumor organoids.** We next aimed to validate the biological relevance of YAP/TAZ for the survival of tumor cells. For this, we treated established

tumor ($n = 6$) and normal ($n = 3$) colon organoids derived from nine independent CRC patients with verteporfin (VP), a YAP/TAZ-specific inhibitor that blocks their interaction with the transcription factor TEAD[30,31]. Following exposure to VP for 48 h, tumor organoids showed markedly suppressed growth and extensive cell death compared to control (DMSO)-treated tumor organoids (Fig. 4i, right panel) as well as VP-treated normal

**Fig. 3 Differentially activated enhancers in human CRC. a** Unsupervised clustering analysis and Pearson correlation heatmap of H3K27ac ChIP-seq data for the 33,131 ChromHMM-defined active enhancers clearly distinguish patient-derived organoids (PDOs) from normal colon tissues. **b** Volcano plot of differentially enriched enhancer regions between PDOs and normal colon mucosa. Dotted lines indicate thresholds for two-sided adjusted *P*-value < 0.01 and |log2 fold-change > 2 (Wald test with Benjamini–Hochberg false discovery rate correction). **c** Percentage of gained enhancers shared by different PDOs (see Supplementary Fig. 4b, c). **d–e** Representative tracks of H3K27ac and ChromHMM profiles, illustrating examples of gained (**d**) and lost (**e**) enhancer regions in PDOs compared to normal colon mucosa. Shaded boxes indicate the presence or absence of H3K27ac peaks. Interactions between gained enhancers and promoter regions based on chromosome conformation capture (capture Hi-C) data on human colon cancer[22] are shown below the graph. **f** The Hippo signaling pathway is the most significantly enriched pathway related to the gained enhancer-associated genes that are upregulated in PDOs compared to normal tissues. The size of the circles corresponds to the number of gained-enhancer associated genes present in the geneset of a particular KEGG pathway (Gene Ratio). The dotted line indicates the threshold for significantly enriched pathways based on a two-sided Fisher's exact test with a false discovery rate (FDR) < 0.05. **g** Visualization of pathway network for tumor-upregulated genes annotated to gained enhancers based on functional enrichment analysis. Pathway terms are represented by circles, the size of which is proportional to the number of genes. The circles are colored according to the enrichment *P*-value based on a two-sided Fisher's exact test.

organoids (Fig. 4i, left panel). Consistently, cell viability in tumor organoids treated with VP was reduced up to 80% of control-treated tumor organoids and was also significantly reduced (*P* = 0.012; one-sided Mann–Whitney *U* test exact *P*-value) compared to VP-treated normal organoids (Fig. 4j and Supplementary Fig. 5g). This is in agreement with the dispensability of YAP/TAZ for normal physiology of the adult intestinal epithelium[28]. These analyses strongly suggest that YAP/TAZ activation is not only a hallmark of cancer initiation regulating the conserved enhancerome landscape, but it is also essential for tumor maintenance.

**Conserved CRC enhancerome is shared by diverse cancer types**. We next asked whether the YAP/TAZ-regulated enhancers identified in CRC represent a conserved feature of epigenetic deregulation in human cancer pathology. For this, we first assessed the chromatin accessibility levels of the highly conserved gained CRC-enhancers that are bound by YAP/TAZ (*n* = 195; Fig. 4g and Supplementary Data 8) in 23 diverse cancer types using ATAC-seq data obtained from TCGA[17] (Fig. 5a). The 195 enhancer regions displayed a strong chromatin accessibility profile across all COAD TCGA samples (Fig. 5a), validating their regulatory role and underlining the CRC-specific nature of the TAZ-regulated conserved enhancerome detected in the PDO library. Notably, 46 out of the 195 active regulatory elements (23%) were accessible in all cancer types (Fig. 5a, cluster in magenta). We refer to these accessible and shared regulatory elements as "ultra-conserved" providing a core of potential pan-cancer enhancers that may be involved in the molecular mechanisms at the basis of tumor biology and maintenance. To validate the deregulation of these core enhancers, we examined the H3K27ac enrichment across different primary tumors and normal tissues. We found that the H3K27ac signal distribution at the pan-cancer enhancers is significantly higher in primary tumors (\*\*\*\**P* < 0.0001, Wilcoxon rank sum test; Fig. 5b; Supplementary Data File 9), confirming the epigenetic rewiring of these enhancer regions in diverse cancer types.

We next assessed the downstream effects of YAP/TAZ regulation by identifying the target genes of the YAP/TAZ-controlled enhancers. Focusing on those target genes that are significantly more expressed in PDOs compared to normal tissues, we confirmed that the majority of them are also upregulated in the tumor samples of the COAD dataset (Supplementary Fig. 6a). Among the genes annotated to the pan-cancer core enhancers, the vast majority (80%) are also directly regulated by the binding of TAZ on their promoters. Some of these genes are oncogenes (i.e., *MYC*) or known targets of YAP/TAZ (i.e., *EREG*, *PHLDA1*, *FJX1*)[28], whereas for other genes (i.e., *UBE2H*) there are no previous reports of their role as YAP/TAZ target genes. The intronic enhancer associated to *TAZ* (Fig. 5c, boxed area) was among the 46 pan-cancer active regulatory regions bound by

YAP/TAZ. Indeed, PDOs, but not normal tissues, consistently display H3K27ac enrichment in this region that coincides with open chromatin or active H3K27ac profile in the vast majority of tumors, suggesting the functional role of this transcriptional activator in diverse human cancer types.

Overall, interrogation of chromatin accessibility data (TCGA) confirms that the YAP/TAZ-regulated CRC enhancerome is active in all the COAD samples. Moreover, extending the significance of our findings to other cancer types, we reveal that a pan-cancer core of enhancers displays an active and highly conserved chromatin profile in several solid tumors, suggesting a role for YAP/TAZ at the roots of deregulated gene expression of the human cancer enhancerome.

**Single-cell landscape of the pan-cancer core of enhancers**. We next sought to depict the cancer epigenetic deregulation at single-cell resolution. To this end, we assessed the cellular distribution and expression of the genes associated with the pan-cancer core of enhancers using single-cell RNA-seq data of primary CRC tissues[32]. We analyzed a total of 63,689 cells from tumor and normal colon tissues of 23 patients comprising diverse tumoral and normal cell populations of epithelial, stromal (endothelial cells; fibroblasts), and immune cells (B and T lymphocytes; myeloid cells; mast cells) that offer a global cellular landscape of CRC (Fig. 6a, Supplementary Fig. 6b). This dataset also represents the heterogeneity of CRC with patients classified into different clinical and molecular subtypes (Supplementary Fig. 6b). We then focused on the cellular landscape of the genes annotated to the YAP/TAZ-regulated pan-cancer enhancers (Fig. 5a, Supplementary Data 8) that define a cancer regulatory blueprint. To better understand the signal intensity and distribution of the gene signature, we created a score of the cancer regulatory blueprint (CRB score) considering both the expression level and co-expression of genes within a cell. We found that the CRB score is highly enriched in the malignant epithelial cells but largely absent from both normal epithelial and other stromal/immune cells (Fig. 6b, c).

YAP/TAZ activation in tumor cells, amongst a diverse array of downstream effects, promotes stem-like properties[26,33]. We thus asked whether the epigenetically driven deregulation could relate to stemness. To obtain a better resolution, we performed subclustering analysis of normal and tumor epithelial cells (Supplementary Fig. 6c) and found that malignant cells with an active CRB do not necessarily display stem-like properties (Fig. 6d–f, Supplementary Fig. 6d). This suggests that the blueprint is a feature of cancer that relates to YAP/TAZ-driven effects on tumoral transcriptional states that reach beyond the acquisition of stemness.

Collectively, we show that the cancer regulatory blueprint is enriched in the malignant cell populations despite the genetic and transcriptional heterogeneity of primary tumors and is associated

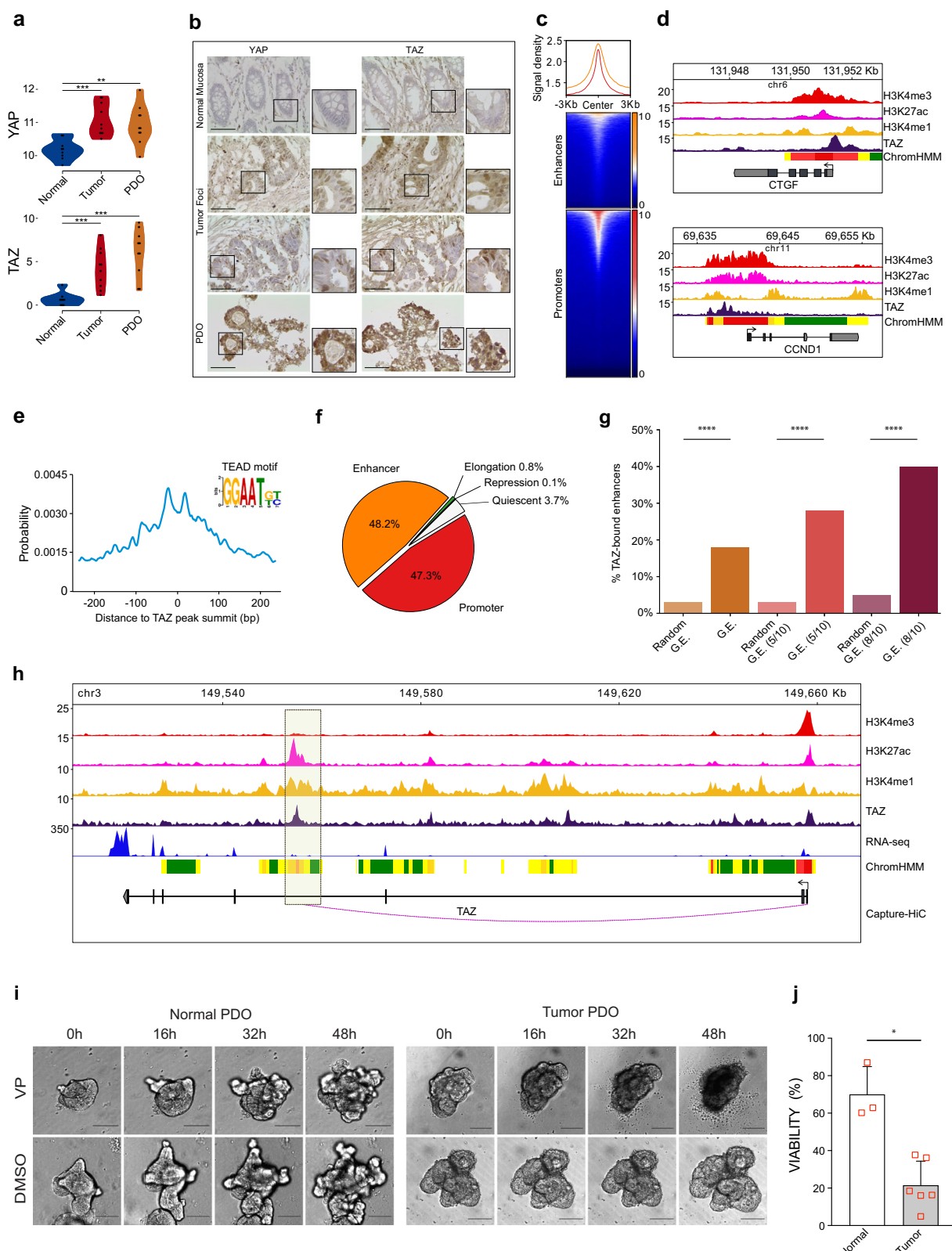

to an aberrant YAP/TAZ activation that is not related to stemness per se.

## Discussion

In our study, the epigenetic landscape of human CRC unveils the existence of an aberrant pan-cancer core of enhancers regulated by the transcriptional coactivators YAP/TAZ and active in more than 20 types of human malignancies. The exploitation of primary tissue-derived tumor organoids, allowed us to dissect cancer cell-intrinsic epigenetic alterations devoid of the influence of stromal cells, and provided sufficient material to perform a systematic de novo discovery of biologically informative chromatin states. By combining more than 66 chromatin maps for multiple

**Fig. 4 YAP/TAZ regulates the conserved human CRC enhancerome. a** *YAP* and *TAZ* are transcriptionally upregulated in primary tumors ($n = 10$) and patient-derived organoids (PDOs $n = 10$) compared to normal colon tissues ($n = 9$). Violin plots of log2 normalized gene counts. **$P < 0.01$, ***$P < 0.001$, two-sided Wilcoxon rank sum exact test. **b** Representative immunohistochemistry images with insets showing YAP/TAZ nuclear localization in tumor tissues (middle) and PDOs (bottom) compared to normal mucosa (top). Scale bars, 50 μm, magnification ×40. **c** Relative distribution of TAZ peaks around ChromHMM-defined active enhancers ($n = 33,131$, yellow) and promoters (red). **d** Genomic overview of YAP/TAZ canonical targets showing H3K4me3, H3K27ac, H3K4me1, TAZ profiles, and ChromHMM states (see Supplementary Fig. 5b and Fig. 2c, e for details on ChromHMM tracks). **e** TEAD binding motif **e**nrichment around the summit of TAZ peaks. **f** Distribution of TAZ peaks across ChromHMM-defined functional elements. **g** TAZ enrichment in human CRC gained enhancers (G.E.) increases with the level of enhancer conservation across PDOs. The bar plots show the percentage of enhancers in each G.E. subset that overlap a TAZ peak or the percentage (mean ± s.d.) of TAZ-bound regions in 1000 random sets generated for each G.E. subsets: (i) all G.E., (ii) G.E. conserved in >5 patients, and (iii) G.E. conserved in >8 patients (see "Methods"). ***$P < 0.001$, one-sided empirical $P$-value. **h** *TAZ* genomic overview showing H3K4me3, H3K27ac, H3K4me1, and TAZ profiles, RNA-seq signals, ChromHMM states, and capture Hi-C promoter-enhancer interactions. **i** YAP/TAZ-inhibition affects patient-derived tumor organoid growth. Representative images of fully formed normal (left) and tumor (right) organoids cultured for 48 h following treatment with 1 μM of verteporfin (VP; top) or DMSO (bottom). Scale bar: 100 μm, magnification ×10. **j** VP treatment significantly reduces cell viability in tumor compared to normal PDOs. Viability was assessed by flow cytometry 48 h after exposure and normalized to DMSO (mean ± s.d., *n*: Normal = 3 and Tumor = 6 independent CRC patients; see Supplementary Fig. 5g). *$P = 0.012$, one-sided Mann–Whitney *U* exact test.

histone marks we identified 8 different epigenetic states (Fig. 2c), revealing the genome-wide location of promoter and enhancer regions, as well as elongating and repressed genomic regions. This data generates an extensive functional annotation of the human genome in CRC allowing the interrogation of diverse modes of epigenetic regulation. These are complemented by matched transcriptomic profiles providing a comprehensive view of correlated activity patterns in human CRC and an essential resource for exploring not only the private epigenetic programs that drive patient heterogeneity but also the common epigenetic blueprints. To facilitate the public use of these omics resources we have created *HePIC*, a web application tool (publicly available at https://bioinformatics.ifom.eu/hepic/) that allows the interactive visualization of both epigenomic (ChIP-seq on histone marks and ChromHMM tracks) and transcriptomic (RNA-seq) data for all the PDOs analyzed in this study.

Epigenomic events and post-transcriptional processes[34] are involved in intestinal homeostasis and colon cancer. Among the epigenetically controlled genomic elements, enhancers are attracting considerable attention due to their fundamental role in tumor development, progression, and metastasis[6,7]. We used chromatin states to provide a more robust annotation of the different classes of enhancers compared to predictions based on ATAC-seq data[17] or individual histone marks[35,36]. As a result, we were able to discriminate active enhancers from other genomic elements within chromatin accessible regions, as exemplified by the epigenetic profiling of the *TAZ* gene locus (Fig. 5c), where multiple open chromatin regions across diverse tumor types correspond to only a handful of active enhancers. These findings are in line with reports by us and others[37] that less than a third of all ATAC-seq identified regions intersect with ChromHMM-defined enhancers, with merely 17% of open chromatin regions for COAD qualifying as active enhancers.

Human CRCs are characterized by ostensibly endless combinations of oncogenic lesions resulting in a high degree of intratumoral and intertumoral genetic heterogeneity[38]. Is the epigenetic level similarly complex or, rather, does it represent a much-simplified layer of integration of genetic and microenvironmental inputs into a restricted, shared set of transcriptional states? Based on ChromHMM-defined chromatin states, we found two main groups of enhancers that are differentially active in PDOs compared to normal mucosa. While half of these enhancers displayed low levels of conservation across PDOs, the remaining half was conserved in at least 50% of the tumor organoids (Fig. 3c), including those displaying microsatellite instability (Fig. 1b). This comes in striking contrast with the reported recurrence of mutated genes in CRC; with the exception

of few driver genes the vast majority of the recurrently mutated genes are shared by <10% of tumors[38]. Thus, our findings indicate that despite the profound genetic heterogeneity, CRC is characterized by a common aberrant enhancerome.

To provide an in-depth characterization of the shared CRC enhancerome, we sought to understand which transcription factors orchestrate the activation of these *cis*-regulatory elements. Although multiple factors may facilitate cell deregulation in CRC, our motif discovery and functional enrichment analyses highlighted the AP1 and TEAD families along with the Hippo pathway, pinpointing the YAP/TAZ transcriptional coactivators as major regulators of the human CRC enhancerome. In line with their pervasive activation in human epithelial tumors[23], we confirmed the transcriptional upregulation and nuclear translocation of YAP/TAZ, and further unveiled the YAP/TAZ chromatin recruitment at distal enhancers in human CRC. In addition, we found an enrichment of TEAD and AP1 motifs at YAP/TAZ-bound genomic elements, supporting the involvement of these TFs as YAP/TAZ partners also in this type of cancer, as suggested by previous studies[21,39].

Strikingly, YAP/TAZ were most enriched in the highly conserved gained enhancers (Fig. 4g), highlighting these transcription factors as driving forces of the shared CRC deregulated enhancerome. The relevance of this epigenetic signature was extended to diverse malignancies of epithelial cells suggesting a previously undescribed universal role for YAP/TAZ as master regulators of tumor-associated epigenetic shifts. In light of the recently reported YAP/TAZ-dependent transcriptional addiction in cancer[10], we speculate that the core of 46 pan-cancer gained enhancers identified in our study could be at the root of the cancer transcriptional addiction, representing a unique epigenetic "fil rouge".

Still, there are unaddressed questions regarding the regulation of YAP/TAZ. Interestingly, we found a tumor-specific epigenetic mechanism of *TAZ* auto-regulation: a positive feedback loop between an intronic YAP/TAZ-bound active enhancer and *TAZ* promoter itself, shared by all CRC PDOs. This intronic enhancer was also observed in the TCGA panel of 23 tumor types, suggesting that the *TAZ* self-regulation is relevant to a wide range of cancers (Fig. 5c). This transcriptional feedback mechanism combined with the inhibition and persistent activation of the Hippo and Wnt pathways, respectively[40], may provide a constant fuel of YAP/TAZ for uncontrolled proliferation of tumor cells, sustaining the YAP/TAZ-dependent transcriptional addiction in cancer[10]. YAP/TAZ activation can be influenced by heterogenous cancer-cell intrinsic features (i.e., underlying genetic lesions) and mechanical cues, and is interconnected with mitogenic signals

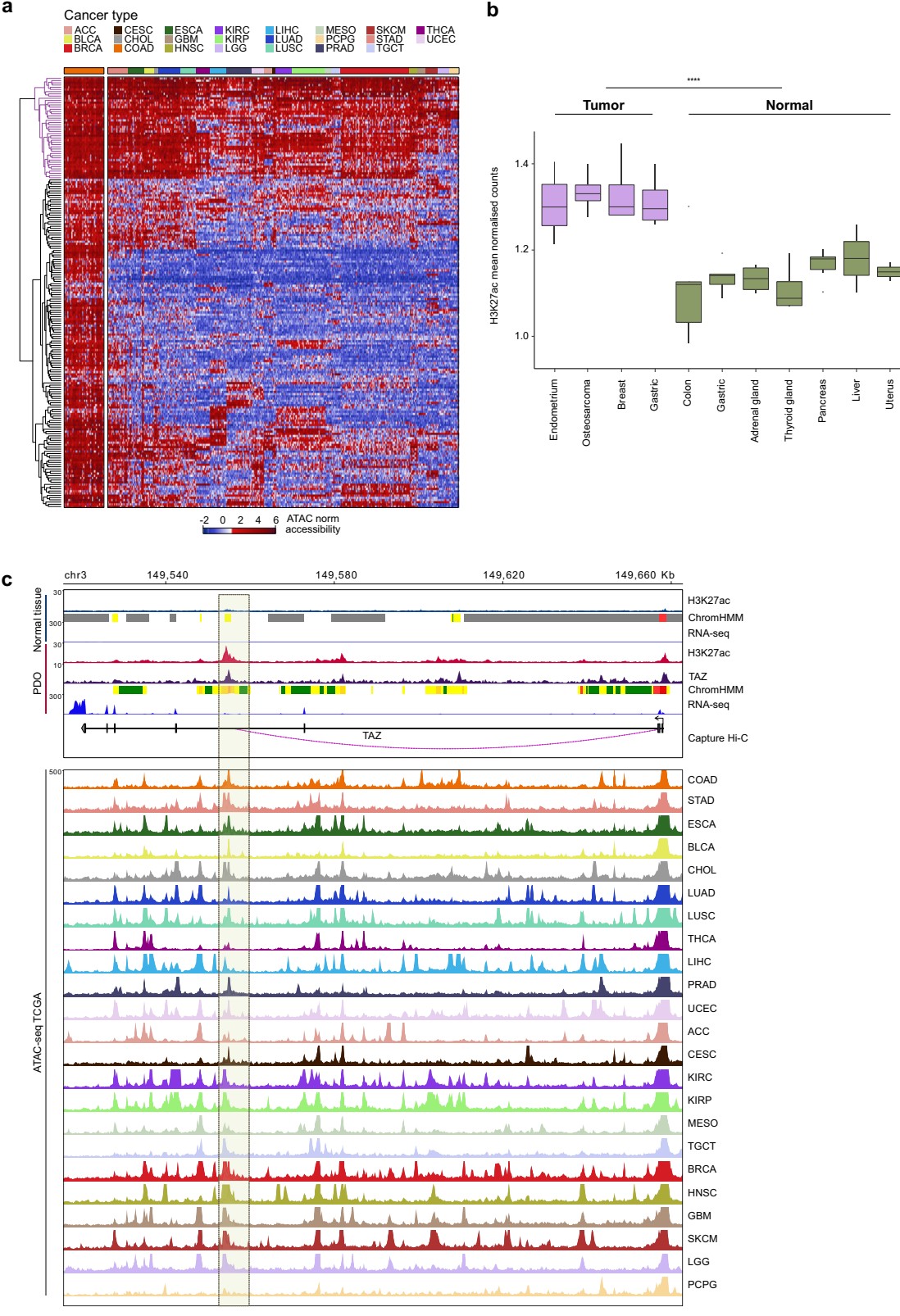

(i.e., MYC) and WNT/β-catenin signaling[28,33,41]. Nevertheless, the YAP/TAZ-mediated addiction concurs with our observations at the single-cell level that the epigenetic deregulation is shared by malignant cell populations of diverse CRC subtypes (Fig. 6d and Supplementary Fig. 6d). In support, our experimental studies demonstrate that inhibition of YAP/TAZ in fully established

human CRC organoids causes growth defects and extensive cell death. Overall, these findings suggest a key role for YAP/TAZ in sustaining a core gene-regulatory network intrinsic to CRC and possibly other malignancies, that is essential for the maintenance of the cancer cell state; a largely unexplored biological property of YAP/TAZ.

**Fig. 5 Conserved human CRC enhancer regions are accessible across diverse cancer types. a** Chromatin accessibility profiles of the 195 conserved gained enhancers in 23 diverse primary human cancer types reveal a signature of 46 pan-CRC enhancers with active chromatin profiles across cancer types (cluster in purple). The heatmap shows hierarchical clustering of log2 normalized insertion counts of ATAC-seq data derived from TCGA (Corces et al.[17]). Colon adenocarcinoma samples are the first cancer type reported on the left of the heatmap. The color-coded bar above the heatmap represents the different cancer types: ACC adrenocortical carcinoma, BLCA bladder urothelial carcinoma, BRCA breast invasive carcinoma, CESC cervical squamous cell carcinoma, and endocervical adenocarcinoma, CHOL cholangiocarcinoma, COAD colon adenocarcinoma, ESCA esophageal carcinoma, GBM glioblastoma multiforme, HNSC head and neck squamous cell carcinoma, KIRC kidney renal clear cell carcinoma, KIRP kidney renal papillary cell carcinoma, LGG brain lower grade glioma, LIHC liver hepatocellular carcinoma, LUAD lung adenocarcinoma, LUSC lung squamous cell carcinoma, MESO mesothelioma, PCPG pheochromocytoma and paraganglioma, PRAD prostate adenocarcinoma, SKCM skin cutaneous melanoma, STAD stomach adenocarcinoma, TGCT testicular germ cell tumors, THCA thyroid carcinoma, UCEC uterine corpus endometrial carcinoma. **b** Differences in the H3K27ac intensities of the pan-cancer enhancers in primary tumors ($n = 28$) relative to normal tissues ($n = 15$) from public ChIP-seq data. Boxplots describe the median (middle line) and interquartile range (box denoting first and third percentile) with whiskers denoting the minimum and maximum within the 1.5× interquartile range and outlying points beyond the whiskers plotted individually. ****$P < 0.0001$, two-sided Wilcoxon rank sum exact test. **c** Genomic overview of the *TAZ* locus in representative normal tissue and patient-derived organoid (PDO) samples, and in TCGA cancer types. Upper panel: H3K27ac profiles, ChromHMM states and RNA-seq signals in normal tissue; H3K27ac and TAZ profiles, ChromHMM states, and RNA-seq signals in PDO; and CRC capture Hi-C data. Bottom panel: ATAC-seq profiles for 23 TCGA cancer types. The shaded box indicates a CRC-conserved and YAP/TAZ-bound ChromHMM-defined active enhancer for which a promoter-enhancer interaction is reported using CRC capture Hi-C data. See Fig. 2c, e for details on ChromHMM tracks.

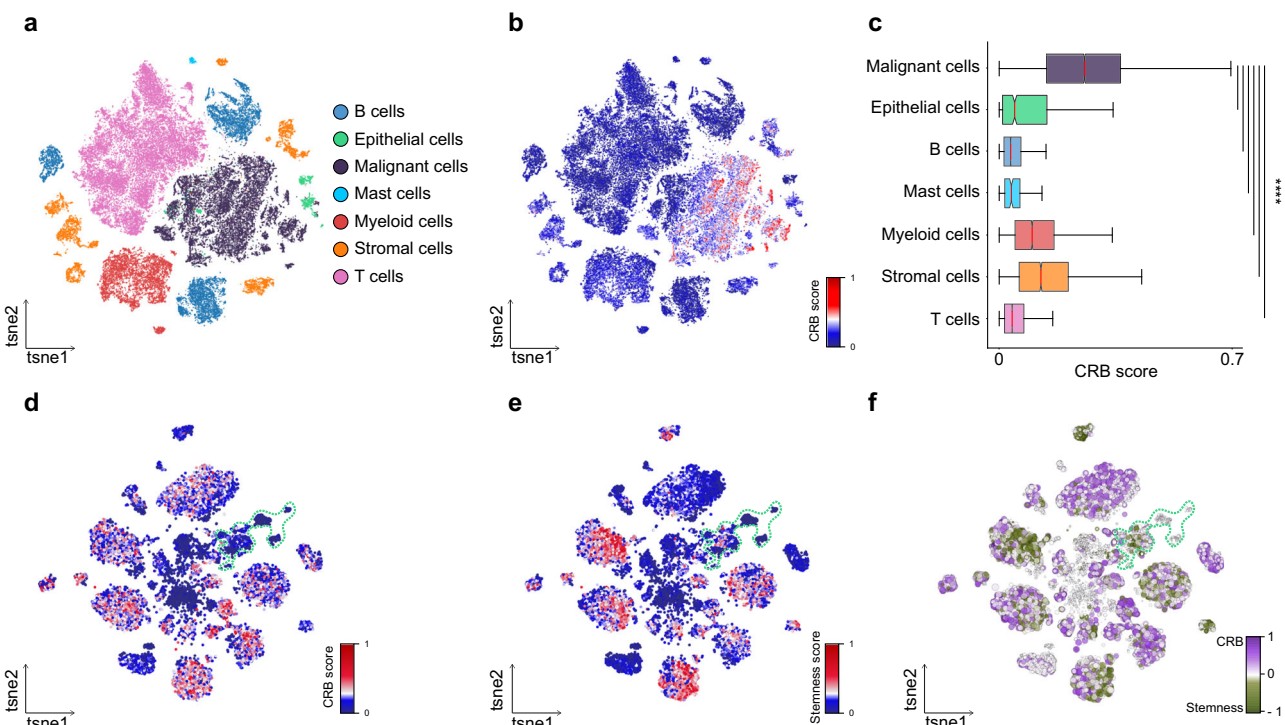

**Fig. 6 Distribution of cancer epigenetic deregulation at single-cell level. a** *t*-distributed stochastic neighbor embedding (t-SNE) visualization depicting the major cell types identified in scRNA-seq data of primary tumor and normal tissues from 23 CRC patients. **b** t-SNE visualization of the CRB scores across all cell populations. **c** Malignant cells display significantly higher CRB scores. Distribution of the CRB score in the malignant (gray; $n = 17,469$) and non-malignant epithelial clusters (green; $n = 1070$), and in all other major cell populations (B cells $n = 9146$; Mast cells $n = 187$; Myeloid cells $n = 6769$; Stromal cells $n = 5933$; T cells $n = 23,115$). Boxplots describe the median (middle line) and interquartile range (box denoting first and third percentile) with whiskers denoting the minimum and maximum within the 1.5× interquartile range. ****$P < 0.0001$, two-sided Mann–Whitney $U$ test. **d–e** t-SNE representation of the CRB (**d**) and stemness (**e**) scores across 18,539 epithelial cells. Contour lines denote normal epithelial cells. **f** The cancer regulatory blueprint does not relate to stemness. t-SNE representation of the difference between the CRB and stemness score across all epithelial cells. Cells depicted in gray display similar levels of the two scores, whereas cells in purple and green display mutually exclusive high levels of CRB or stemness, respectively. Contour lines denote normal epithelial cells.

Deeper understanding of the mechanisms by which YAP/TAZ exert their nuclear functions can link the inhibition of these coactivators to an arsenal of potent epigenetic agents. For instance, YAP/TAZ-mediated transcriptional addiction is achieved through interaction with the bromodomain and extraterminal domain (BET) coactivator BRD4, demonstrating the rational use of BET inhibitors in impairing the expression of YAP/TAZ-regulated genes and YAP/TAZ-induced oncogenic functions and drug resistance[10]. A different approach to control YAP/TAZ activity could arise from the epigenetic modulation of *TAZ* gene expression. The dispensability of YAP/TAZ for normal tissue homeostasis[26,28], also confirmed by us, provides a further argument in favor of exploiting YAP/TAZ as master regulators of a cancer regulatory blueprint to discover potential targets for epigenetic-based therapeutic approaches.

## Methods

**Human specimens.** Primary CRC tissues were obtained from San Gerardo Hospital (Department of Surgery), Monza and UO Chirurgia Epatobiliopancreatica e Digestiva Ospedale San Paolo, Milan following ethical approval from their Institutional Review Boards. Written informed consent was obtained from all patients for the usage of their samples for research purposes, including the creation of PDOs. Clinical details on patients are provided in Supplementary Table 1. Samples were confirmed to be tumor or normal based on pathologist assessment and were obtained prior to treatment. MSI-MSS status was determined according to standard experimental procedures[42].

**Isolation of human primary tissues.** Primary colonic normal and tumoral tissues were processed as described below[43]. Surgically resected specimens were reduced to a size of 3–5 mm, extensively washed with cold PBS and gentamicin (20 μg/ml) and incubated with PBS-EDTA (2.5 mM) rocking on a wheel for 1 h at 4 °C. After PBS-EDTA treatment, tissue samples were washed with cold PBS—1% FBS to release normal crypts and tumoral counterpart. Cells suspension were collected by centrifuging at $400 \times g$ for 5 min at 4 °C and used for transcriptomic and epigenomic analyses.

**CRC patient-derived organoid libraries.** CRC and normal colon PDOs were established and maintained as described below[43]. Tumor and normal cell suspensions isolated from surgical resections were embedded in drops of Matrigel® Growth Factor Reduced Basement Membrane Matrix, Phenol Red-Free (Corning) to establish CR/CRC PDOs libraries. Droplets of matrigel containing tumor/normal cells suspension or established organoids were maintained in 24-well plates overlaid by 500 μl of the organoid culture medium (Advanced DMEM/F12 (Life Technologies) supplemented with penicillin/streptomycin (Euroclone), 10 mM HEPES (Life Technologies), 2 mM GlutaMAX (Life Technologies), 1X B27 (Life Technologies), 1X N2 (Life Technologies), 1 mM N-Acetyl Cysteine (Sigma-Aldrich), 10 mM Nicotinamide (Sigma-Aldrich), 50 ng/ml human EGF (Peprotech), 100 ng/ml human Noggin (Peprotech), 10 nM human Gastrin (Sigma), 500 nM A83-01 (Tocris), 10 μM SB202190 (Sigma). The normal colon PDOs were cultured also in the presence of 50% Wnt3a-conditioned media (stably cell line produced in-house) and 1 μg/ml human RSPO1 (Peprotech, 120-38). The organoids were split once per week by mechanical disruption or enzymatic digestion using TrypLE™ Express Enzyme (Thermo Fisher, 12605010) and regularly checked for mycoplasma contamination. For the VP experiment, the normal/tumoral PDO lines were treated with 1 μM VERTEPORFIN (VP - Sigma-Aldrich, SML0534) or its vehicle dimethyl sulfoxide (DMSO - Sigma-Aldrich, D2650) as control for 48 h.

**Organoid immunofluorescence and microscopy.** For whole-mount staining[44], isolated organoids embedded in Matrigel in μ-Plate Angiogenesis 96 Well (Ibidi), were fixed in 4% paraformaldehyde in PBS for 1 h, at 4 °C. After fixation, the autofluorescence was quenched with 50 mM NH4Cl for 30 min and the organoids were permeabilized with 0,5% Triton X-100 (Sigma-Aldrich) for 1 h and blocked with 10% Donkey Serum or Normal Goat Serum (Sigma-Aldrich) in PBS with 0.2% Triton X-100 overnight, at 4 °C in mild shaking. Primary and secondary antibodies were diluted in 5% of serum and incubated for ~35 and ~12 h, respectively, at 4 °C in mild shaking (Supplementary Table 2). Cell nuclei were stained with 20 μg/ml Hoechst 33342 in PBS with 0.2% Triton X-100 for 2 h and the organoids were stored in PBS with 0.02% NaN2 until the acquisition. Fluorescence images were captured with confocal laser-scanning SP5 microscope (Leica Microsystems) equipped with eight laser lines and four PMT detectors, using 10× (NA 0.3) or 20× (NA 0.7) dry objectives (TCS SP5; Leica), 5 or 10 μm z-step interval and 1024 × 1024 or 2048 × 2048 image format. For each acquired confocal z-stack field, maximum intensity projections (MIP) were generated using ImageJ software (National Institutes of Health). For YAP/TAZ-inhibition experiment, normal and tumoral PDOs were seeded in 96-well plate at the same confluence and followed in live imaging for 48 h at Leica Thunder Imaging System. Each brightfield (BF) image was captured every 16 h using 20 μm Z-steps and was processed for tiling and extended depth of field (EDF). Further image analysis was performed using Fiji software (v2.1.0/1.53c) (National Institutes of Health).

**Immunohistochemistry and in situ hybridization.** Immunohistochemical staining of human CRC primary tissues and PDOs was performed on formalin-fixed, paraffin-embedded, or on fresh OCT-embedded tissue and PDO sections (Fig. 4b and Supplementary Fig. 5f). For immunohistochemistry, frozen samples were cryostat sectioned, fixed in 100% cold acetone for 2 min, blocked with 2% FBS serum in PBS for 60 min, and incubated overnight with primary antibodies (WWTR1/TAZ 1:20 Sigma-Aldrich, HPA007415; YAP 1:100, Santa Cruz Biotechnology – Sc-101199). The antibodies were detected using a polymer detection kit (GAM/R-HRP, Microtech) followed by a diaminobenzidine chromogen reaction (Peroxidase substrate kit, DAB, SK-4100; Vector Lab). All sections were counterstained with Harris's Hematoxylin and visualized using a bright-field microscope. For in situ hybridization, tissue sections (formalin fixed, paraffin embedded) were processed for RNA in situ detection using the RNAscope Duplex Detection Kit (Chromogenic) according to the manufacturer's instructions

(Advanced Cell Diagnostics). RNAscope probe was *FOXQ1* (NM_033260.3, region 694 - 2197), which was detected using the HRP-based Green detection reagent.

**Flow cytometry.** Normal and tumoral PDOs treated with VP or DMSO were collected and dissociated to single-cell level using TryPLE express solution, for 5 min at 37 °C. Cells were stained with Fixable Viability Stain 780 (FVS780 - BD HORIZON™, 565388) according to manufacturer's instructions and samples were acquired at FACSCanto II (BD), following the manufacturer's instructions (Supplementary Fig. 5h for gating strategy).

**RNA isolation and bulk RNA-seq library construction.** For RNA-seq analysis, cell suspensions from CRC PDOs, primary normal, and tumor tissues were lysed in TRIzol reagent (Thermo Fisher) and processed for total RNA extraction with PureLink™ RNA Mini Kit (Thermo Fisher), according to manufacturer's instructions. PDOs samples were collected at early (<5 splits) and late passages (>5 splits). The RNA quality was assessed by the RNA Integrity Number (RIN) value with RNA6000 assay (Agilent). Only samples with RIN > 7.0 were used in this study. RNA-seq libraries were constructed according to the TruSeq mRNA Stranded preparation kit (Illumina, San Diego, USA) and sequenced at HiSeq2500.

**Chromatin immunoprecipitation (ChIP) assay.** For ChIP experiments, matrigel droplet containing ~0.3 × 10⁶ cells/well was dissolved using Cell Recovery Solution (Matrisperse Cell Recovery Solution - Sacco-L004419 CPB40253), following the indicated procedure. PBS-washed organoids pellet was fixed as a whole in Formaldehyde (F8775 SIGMA) PBS-solution (final 1%), for 10 min rocking at room temperature and quenched with 0.125 M Glycine for 5 min. PBS-washed organoid pellets were lysed in 500 μl of 1× sonication lysis buffer (10 mM Tris pH 8.0, 0.25% SDS, 2 mM EDTA, plus protease inhibitors) and incubated for at least 10 min at 4 °C. Lysed chromatin was sheared at 200–500 bp fragments using Covaris® M220 focused-ultrasonicator (settings: duty factor 20%, peak incidence power 75 Watt, cycles per burst 200, 8–15 min). For organoids and primary tissues ~500 ng and ~1000 ng respectively of the sonicated chromatin was incubated with 0.5/1 μg of histone mark antibodies (H3K27Ac abcam 4729; H3K4me3 Millipore 07-473; H3K4Me1 DIAGENODE C15410194; H3K36me3 DIAGENODE C15410192; H3K27me3 07449 Millipore), overnight at 4 °C. Immunocomplexes were recovered the following day with blocked 10 μl Protein G-Dynabeads (Thermo Fisher) for 2 h at 4 °C and washed twice with RIPA-low salt (10 mM Tris-HCl pH 8.0, 140 mM NaCl, 1 mM EDTA pH 8.0, 0.1% SDS and 1% Na-Deoxycholate, 1% Triton x-100), twice with RIPA-high salt (10 mM Tris-HCl pH 8.0, 1 mM EDTA pH 8.0, 500 mM NaCl, 1% Triton X-100, 0.1% SDS, 0.1% DOC), twice with RIPA-LiCl (10 mM Tris-HCl, pH 8.0, 1 mM EDTA, pH 8.0, 250 mM LiCl, 0.5% NP-40, 0.5% DOC), once with 10 mM Tris pH 8.0 and once with 1X TE, followed by reverse crosslinking overnight. The washed immunocomplexes were incubated with ChIP elution buffer (10 mM Tris-HCl pH 8.0, 5 mM EDTA pH 8.0, 300 mM NaCl, 0.4% SDS) supplemented with 0.8 mg/ml Proteinase K for 1 h at 55 °C and overnight at 65 °C, for reverse crosslinking. The immunoprecipitated DNA was then purified by Qiagen MinElute kit (Qiagen) and eluted in 22 μl EB buffer. ChIPmentation on YAP/TAZ[45] was carried out, with the following modifications. Briefly, crosslinked organoids were lysed in buffer I (50 mM HEPES, pH 7.5, 10 mM NaCl, 1 mM EDTA, 10% Glycerol, 0.5% NP-0.4, 0.25% Triton X-100, plus protease inhibitors) in ice. Organoids were recovered and lysed in buffer II (10 mM Tris-HCl pH 8.0, 200 mM NaCl, 1 mM EDTA, 0.5 mM EGTA, plus protease inhibitors) at room temperature. Organoids pellet was sonicated in lysis buffer III (10 mM Tris-HCl pH 8.0, 200 mM NaCl, 1 mM EDTA, 0.5 mM EGTA, 0.1% Na-deoxyxholate, 0.5% N-lauroysarcosine, plus protease inhibitors) using Covaris® M220 focused-ultrasonicator. Sonicated chromatin was incubated with anti-WWTR1 (Sigma Aldrich, HPA007415), anti-YAP1 (abcam 52771), or anti-rabbit IgG (SinoBiological CR1) overnight at 4 °C on the wheel. Antibody/antigen complexes were recovered with blocked Protein G-Dynabeads (Life Tecnologies) and washed with low salt wash buffer ((20 mM Tris-HCl pH 8.0, 150 mM NaCl, 0.1% SDS, 1% Triton X-100 and 2 mM EDTA), high salt buffer (20 mM Tris-HCl pH 8.0, 2 mM EDTA, 500 mM NaCl, 0.1% SDS, and 1% TritonX) and with 10 mM Tris pH 8.0. Washed immunocomplexes were resuspended in tagmentation reaction (Nextera DNA Sample Prep Kit (Illumina), as described in Schimdl et al.[45]. Beads were washed with low-salt buffer, 1X TE and incubated with ChIP elution buffer (10 mM Tris pH 8.0; 5 mM EDTA; 300 mM NaCl; 0.4% SDS) plus 0.8 mg/ml Proteinase K (NEB) for 1 h at 55 °C and 8 h at 65 °C, to revert formaldehyde crosslinking. The immunoprecipitated and input DNA were then purified by Qiagen MinElute kit (Qiagen) and eluted in 22 μl EB buffer.

**ChIP-seq library preparation.** ChIP-seq libraries were constructed with TruSeq ChIP Library Preparation Kit (Illumina), according to the manufacturer's instructions and sequenced on Illumina HiSeq2500 platform. Library preparation for ChIPmentation was performed using Nextera primers[45] (Supplementary Table 3) and enriched libraries were purified using 1.8 V of SPRI AMPure XP beads and sequenced with Illumina HiSeq2500.

**RNA-seq QC and data analyses.** Quality control of the reads was performed with FastQC (v0.11.7) and MultiQC (v1.5) (http://www.bioinformatics.babraham.ac.uk/

projects/). The reads were trimmed using BBDuk – BBMap v38.16 (https://jgi.doe.gov/data-and-tools/bbtools/bb-tools-user-guide/bbmap-guide/) (command line parameters: ktrim = r k = 23 mink = 11 hdist = 1 tpe tbo qin = 33) and aligned to the human hg38 reference (GENCODE Release 25 basic gene annotation) using STAR (v2.5.3a)[46] (command line parameters: --utFilterMismatchNmax 9 --outFilterMultimapNmax 20 -- alignSJoverhangMin 8 --alignSJDBoverhangMin 1). Quantification was performed using featureCounts[47] – Subread v1.6.2 with default parameters. RNA-seq and ChIP-seq primary analyses were executed by a custom pipeline managed by Nextflow (v20.04.1.5335)[48].

The raw count matrix containing all the samples was created using a custom bash script. Mitochondrial genes were removed from downstream analyses. Normalization and differential analysis were carried out using DESeq2 package (v1.22.2)[49] and R (v3.5.1)[50]. Sample variance for each gene was calculated using the *rowVars* function of R across all samples. PCA was performed using the R function *prcomp* considering the 500 most variable transcripts with parameters center = TRUE, scale = TRUE. The 500 most variable genes were manually inspected and immune/stroma infiltrate-related genes were removed (Supplementary Data 2). Genes were considered differentially expressed with a *P* adjusted ≤ 0.01 upon correction for multiple testing using the Benjamini–Hochberg method in DESeq. A single organoid for each patient was chosen for downstream differential expression analyses in order to maintain an even sample size across the three experimental groups. Heatmap and hierarchical clustering were performed using Z-score normalized counts of DEG and the *pheatmap* function with default parameters (clustering_distance_cols = euclidean, clustering_method = complete). GSEA was performed using the GSEAPy (v0.9.9)[51] package utilising the pre-ranked module with default parameters (permutation_num = 1000). For the visualization of RNA-seq tracks, normalized coverage tracks were generated using the bamCoverage (command line --normalizeTo1x 3049315783 --minMappingQuality 10) function of deepTools (v2.5.7)[52]. Separate tracks for forward and reverse transcripts were generated for each independent sample.

**Tumor primary tissue classification**. The classification of CRC primary tissues was based on the Consensus Molecular Subtype[2], the CRC intrinsic subtypes[3], and the CRCassigner[14] classification systems. The classification was performed using the CMScaller[53] R package (https://github.com/peterawe/CMScaller) with default parameters (false discovery rate (FDR) = 0.5, seed = 1).

**ChIP-seq data processing**. Quality control of the reads was performed with FastQC v0.11.7 and MultiQC v1.5. Reads were aligned to the human hg38 reference (GENCODE Release 25 basic gene annotation) using Bowtie (v1.2.2)[54] (command line parameters: -m 1 --best --strata -v 3), sorted using SAMtools (v1.8)[55] and directly converted into binary files (BAM). PCR duplicated reads were marked and removed using SAMtools. The peaks were called with MACS2 (v2.1.0)[56] (command line parameters: --nomodel --extsize 200 -B -q 0.01 for sharp histone modifications H3K4me3 and H3K27Ac, and adding --broad for broad histone modifications H3K4me1, H3K36me3, and H3K27me3) using matched input DNA as a control. Peaks overlapping ENCODE blacklisted regions hg38 (i.e., regions in the human genome with signal artefacts in next-generation sequencing experiments, https://www.encodeproject.org/annotations/ENCSR636HFF/) were removed. Peaks found in un-placed and un-localized scaffolds were removed. For the visualization of ChIP-seq tracks, Bedgraph tracks were generated using MACS2 *bdgcmp* function, and converted into bigwig using UCSC *bedClip* and *bedGraphToBigWig* functions. pyGenomicTrack[57] was used for the visualization of the tracks.

Filtered and sorted BAM files were used to generate normalized coverage tracks using the *bamCoverage* function from deepTools suite v2.5.7 (--normalizeTo1x 3049315783 --extendReads 200 --binSize 1). The average signal profile and the heatmap plot along the genebody were calculated using *computeMatrix scale-regions* with default parameters (--regionBodyLength 6000 --upstream 3000 --downstream 3000) and GENCODE Release 25 basic gene annotation.

**Correlation analysis of histone marks**. To obtain the correlation heatmap of all the histone modifications for the ten PDOs, a consensus peakset was generated using DiffBind (v2.6.6)[58] by merging together only peaks detected in at least two tracks. Then, a count matrix of 180,250 peaks × 48 samples was created by counting the number of reads per peak for each sample using the *dba.count* with default parameters. The correlation heatmap and PCA plot were produced using *dba.plotHeatmap* (distMethod = "pearson") and *dba.plotPCA* respectively, with default parameters.

**Analyses of publicly available ChIP-seq datasets**. Publicly available ChIP-seq datasets for normal and tumor colon tissues (GSE77737), and colon cancer cell lines HCT116 and Caco2 (ENCODE) (Supplementary Data File 3) were reanalyzed and processed using the same pipeline described above. Publicly available datasets for H3K27ac ChIP-seq used in Fig. 5b (Supplementary Data File 9) can be found under accession codes GSE51776 (gastric cancer), GSE114737 (breast and endometrial cancer), GSE74230 (osteosarcoma), GSE101065 and GSE142924 (uterus), GSE16256 and GSE96504 (liver), GSE16256, GSE101019, and GSE95966 (adrenal

gland), GSE101258, GSE96258, GSE95981, and GSE142995 (thyroid gland), GSE16256, GSE101269, GSE101231, GSE142968, GSE96212 (pancreas).

**De novo chromatin state characterization**. De novo chromatin state characterization of all PDOs was performed using a multivariate Hidden Markov Model approach implemented in ChromHMM (v1.12)[16] considering five histone modifications (H3K4me3, H3K27ac, H3K4me1, H3K36me3, and H3K27me3) across ten PDOs and including additional public available ChIP-seq data (Supplementary Data 3), using default parameters. The read counts for all the considered samples were computed in non-overlapping 200-bp bins across the entire genome. The presence or absence of each histone mark is determined within each interval based on the observed reads counts relative to a Poisson background distribution, Binarization was performed comparing ChIP-seq read count to corresponding input DNA as a control to reduce the technical noise. Several models were trained in parallel using 8, 10, and 12 states. The 8-state model was chosen for downstream analysis since it captured the key interaction between histone marks with minimal redundancy. Figure 2c reports the histone marks emission probability heatmap which represents the frequency with which different histone modifications are co-present in the same genomic region. The annotation names attributed to the states were defined based on the Roadmap Epigenomics Consortium nomenclature[59]. Briefly, two states were annotated as promoter states ("Flanking Active TSS - FlnkActTSS" and "Active TSS - ActTSS") based on the presence of H3K4me3, or the enrichment of both H3K4me3 and H3K27ac, respectively. The two states with strong enrichment of H3K4me1 and H3K27ac and absence of H3K4me3 were defined as "Flanking Active Enhancers - FlnkActEnh" and "Active Enhancers - ActEnh". The state characterized by the presence of H3K4me1 alone was defined as "Weak Enhancers - WkEnh". The "Elongation – Elong" and "Repression - Repr" states were characterized by the presence of H3K36me3 and H3K27me3, respectively. "Quiescence" state marks regions without any significant enrichment of histone marks.

**Overlap of ChromHMM states and COAD ATAC-seq data**. We estimated the probability of detecting previously reported open chromatin regions for colon cancer within each chromatin state. To this end, ATAC-seq data for COAD was downloaded from the TCGA site (https://gdc.cancer.gov/about-data/publications/ATACseq-AWG/)[17]. The number of ATAC-seq peaks inside each ChromHMM state was defined by overlapping the regions of each ChromHMM state with the ATAC-seq peak summits. Since each ATAC-seq peak was reduced to the summit of the peak, the length of each ATAC-seq peak corresponded to 1 bp. Then, a conditional probability was calculated to estimate the probability of identifying open chromatin regions in each chromatin state across the ten PDOs. The probability $p(A|B)$ is the probability that event A will occur given the knowledge that event B has already occurred. The conditional probability of A given B is defined as the quotient of the probability of the joint event A and B (both events A and B occur together) and the probability of B (1).

$$p(A|B) = \frac{p(A \cap B)}{p(B)} \quad (1)$$

For "*i*" = 1..*n* where *n* is the number of chromatin states, $p(A)$, $p(B)$ and $p(A \cap B)$ were defined as follows:
$p(A)$ = total length of ATAC-seq peaks/total length of the genome
$p(B)$ = total length of ChromHMM$_i$ state/total length of the genome
$p(A \cap B)$ = total length of ATAC-seq peaks overlapping ChromHMM$_i$ state/total length of the genome
With $p(A|B)$ defined as the probability of finding ATAC-seq peaks in each ChromHMM state:
$p(A|B)$ = total length of ATAC-seq peaks overlapping ChromHMM$_i$ state/total length of ChromHMM$_i$ state.

**Identification of active enhancers**. To identify tumor-specific active enhancers, all the "Active Enhancer" and "Flanking Active Enhancers" regions from the ten PDOs and the five normal colon tissue ChromHMM states were selected. These two states are defined by the co-presence of high level of H3K27Ac and H3K4me1 signals. The pool of active enhancer regions was filtered excluding all regions with <200 bp length and within a window of 5 Kb around the TSS (based on known genes within GENCODE Release 25 basic gene annotation). Then, a consensus peakset was built using DiffBind, as described for correlation analyses of histone marks. The number of H3K27ac reads in the consensus peakset was counted generating a count matrix of 33,131 regions × 15 samples. Differential analysis was performed using DESeq2 considering as differentially activated all the regions with a *P* adjusted ≤ 0.01 and a |log2FC| ≥ 2.

**Active enhancer conservation across patients**. A master list of active enhancer regions across all PDOs was produced merging together the "Active Enhancer" and "Flanking Active Enhancers" (Identification of active enhancers) states using BEDTools[60]. Enhancer regions present in at least one patient were considered in this analysis. To assess conservation, the enhancerome of each patient was intersected with the master list of enhancers using a custom script in Python resulting in a matrix that reported the presence ("1") or absence ("0") of each enhancer region

across PDOs. To assess conservation, enhancers were stratified according to their frequency across PDOs and further filtered for enhancers differentially activated (gained) in PDOs compared to normal tissues.

**Motif binding discovery**. We performed motif discovery within the accessible regions of the conserved gained enhancers. First, the ATAC-seq peak set for COAD was downloaded from the TCGA site (https://gdc.cancer.gov/about-data/publications/ATACseq-AWG/). To identify putative open chromatin regions inside the most conserved enhancers, gained enhancer regions, conserved in at least 80% of the patients ($n = 486$), were overlapped with the COAD ATAC-seq peaks. HOMER (v4.7)[61] findMotifsGenome function was used to evaluate the enrichment of known motifs in the exact size of the accessible regions (setting region size parameter to "given") compared to background sequences using the default cumulative binomial distribution to score enrichment. Transcription factor binding motifs encompassing the summit of TAZ peaks were identified with HOMER and MEME-chip[62] on 500 bp windows centered around TAZ peak summits.

**Annotation of differentially activated enhancers**. Differentially activated enhancers were annotated to their putative interacting promoter region using chromosome conformation capture (capture Hi-C) data on human CRC HT29 cell line[22]. For gained enhancers not annotated with capture Hi-C, we selected and merged all regions of the "Active TSS" and "Flanking active TSS" states for the ten PDOs and the five normal tissue. Then, we identified all the protein-coding genes (GENCODE Release 25 basic gene annotation) overlapping with the above active promoter regions. ($n = 13,802$). Finally, we used the *annotatePeakInBatch* (*output* = "both", PeakLocForDistance = "middle") function of ChIPpeakAnno (v3.12.7)[63] to annotate active enhancers to their nearest protein-coding gene with an active promoter.

**Functional enrichment analysis**. We used over-representation analysis based on Fisher's exact test to assess the functional enrichment of biochemical and signaling pathways in the list of 495 tumor-upregulated genes annotated to gained enhancers. Functional enrichment analysis was conducted in R using the *fisher.test* function of the *stats* package on the 321 gene sets of the KEGG collection (downloaded from ConsensusPathDB; http://cpdb.molgen.mpg.de/) considering a genomic background of 21,528 unique gene symbols (given by the union of the 19,950 protein-coding genes of the human hg38 reference GENCODE Release 25 and of the genes of all KEGG gene sets). P-values have been adjusted (i.e., false discovery rate) using the *p.adjust* function of R *stats* package and the threshold for statistical significance set at FDR < 5%. The visualization of the functional enrichment analysis results was obtained in Cytoscape[64] using its EnrichmentMap and AutoAnnotate applications (with default parameters).

**ChIPmentation data processing and QC**. ChIPmentation data was analyzed and processed as described above (ChIP-seq data processing), with the difference that adapters were removed before the alignment of the reads using BBDuk (command line parameters: ktrim = r k = 23 mink = 11 hdist = 1; BBMap v38.16). Peaks were called using MACS2 (v2.1.0)[56], using as control the associated ChIPmentation on the input control (command line parameters: --nomodel --extsize 200 -B -p 0.001). P-values were corrected for multiple comparisons using the Benjamini–Hochberg correction. Density plots and heatmaps were produced as described above (ChIP-seq data processing), with the difference that enrichment was performed on all the promoter regions in GENCODE Release 25 basic gene annotation and the active enhancer regions identified ($n = 33,131$).

**Analysis of TAZ ChIPmentation data**. TAZ peaks were overlapped with the previously defined ChromHMM states using BEDTools, to assess the preferential binding localization of TAZ in (i) all gained enhancers (G.E.) identified in PDOs, (ii) the G.E. conserved in at least five patients, and (iii) those conserved in at least 8 patients. The BEDTools *shuffle* function was used to generate 1000 shuffle tracks separately for each of the above G.E. subsets, preserving the number and size of features in each of the G.E. subsets in the input BED file. Each feature in the input BED file was repositioned in genomic regions of the ChromHMM-defined enhancerome ($n = 33,131$ PDO and normal tissue enhancers), excluding the genomic regions of PDO gained enhancers. In counting TAZ-bound regions, a single count was considered for a region that overlapped multiple TAZ peaks. To assess significance, we computed a one-sided empirical P-value by evaluating the number of times the % of TAZ-bound reshuffled regions in a random set was as extreme as the observed % of TAZ-bound enhancers in a specific G.E. subset, divided by 1000 (number of permutations performed). None of the 1000 reshuffled regions in each of the random sets had a percentage as high or higher than the observed percentage of TAZ-bound enhancer regions in the corresponding G.E. subset.

**Analysis of TCGA pan-cancer ATAC-seq data**. To identify potential pan-cancer regulatory regions, pan-cancer ATAC-seq peak sets from the TCGA consortium were used (https://gdc.cancer.gov/about-data/publications/ATACseq-AWG/)[17]. The pan-cancer peakset was overlapped with the YAP/TAZ-bound gained

enhancers conserved in at least eight patients ($n = 195$). If multiple ATAC-seq peaks were assigned to each enhancer only that with the highest normalized enrichment score was considered. Then, the normalized ATAC-seq insertion counts of the pan-cancer peak sets was downloaded from TCGA site (https://gdc.cancer.gov/about-data/publications/ATACseq-AWG/) and was used to produce a heatmap (*pheatmap;* clustering_distance_cols = euclidean, clustering_method = complete) of all the TCGA patients and the 195 enhancer regions of interest. To identify pan-cancer accessible regions, we performed hierarchical clustering with cluster_rows = TRUE.

**Analysis of publicly available H3K27ac ChIP-seq datasets**. Publicly available ChIP-seq data for H3K27ac were obtained from the Gene Expression Omnibus (Supplementary Data 9). Raw sequencing reads were processed as described above (ChIP-seq data processing). For each sample, the number of H3K27ac reads in the consensus peakset of ~33 K active enhancers (Identification of active enhancers) were counted in DiffBind. Read counts across samples were normalized and corrected for potential batch effects using *ComBat*[65]. For each of the primary tumor and normal tissue samples, the mean H3K27ac normalized counts across all 46 pan-cancer enhancer regions were calculated and the Wilcoxon rank sum test was performed to determine the difference in H3K27ac intensities between primary tumors and normal tissues.

**Analysis of TCGA COAD RNA-seq data**. The gene expression quantification (HTSeq counts) and the clinical information for COAD were downloaded from the R package *TCGAbiolinks*[66] (https://bioconductor.org/packages/release/bioc/html/TCGAbiolinks.html) (*GDCquery*(project = "TCGA-COAD", data.category = "Transcriptome Profiling", data.type = "Gene Expression Quantification", workflow.type = "HTSeq - Counts")). Both normal and tumor tissues were included in the analysis while the Formalin-Fixed Paraffin-Embedded (FFPE) tissue specimens were removed. The RNA-seq counts were normalized using DESeq2 as described above. To retrieve the percentage of epithelial cells in normal and tumor samples, decomposition of the bulk RNA-seq expression data was performed using *BisqueRNA*[67] (https://github.com/cran/BisqueRNA). The decomposition was computed using default parameter and integrating scRNA-seq for 23 CRC primary tissues[32] previously annotated for major cell types (markers = NULL, use.overlap = FALSE). To evaluate the expression level in epithelial cells, the COAD log2 normalized counts for target genes were multiplied for the epithelial cell proportion as determined in BisqueRNA.

**Single-cell RNA-seq data processing of primary CRCs**. The raw count matrix of 3′ end scRNA-seq data (10X technology) on 63,689 cells from 23 CRC patients consisting of 23 primary CRC and 10 matched normal mucosa samples were downloaded from GSE132465. The matrix was then processed using the python package Scanpy (v1.4.2)[68]. First, genes detected in <0.1% of the total cells and cells with fewer than 200 expressed genes were removed. The total count matrix used for downstream analysis was composed of 63,689 cells and 33,694 genes. The matrix was normalised considering a scaling factor of $10^4$ and log-transformed using *scanpy.pp.normalize_per_cell(data, counts_per_cell_after = 1e4)* and *scanpy.pp.log1p(data)*. Highly variable genes (HVG) were selected based on specific thresholds for mean expression and dispersion using *scanpy.pp.highly_variable_genes (min_mean = 0.08, max_mean = 4, min_disp = 0.7)* and excluding mitochondrial and ribosomal genes. The cell cycle phase of each cell was evaluated by scoring individual cells for their expression of G1-, S-, and G2M-phase genes[69].

**Dimensionality reduction and clustering of scRNA-seq data**. PCA was performed on scaled and centered values considering 1093 HVG. Unwanted sources of variation (i.e., number of detected counts and genes per cell, the percentages of mitochondrial and ribosomal counts and the cell cycle phase) were evaluated and regressed out using linear regression as implemented in scanpy (*scanpy.pp.regress_out*). Initially, a K-Nearest Neighbor graph was constructed based on Euclidean distance in PCA space, thus refining the weight of the edges between two cells using Jaccard similarity (*scanpy.pp.neighbors* with n_neighbors = 15, n_pcs = 15). For reproducibility, we used the clustering annotation reported by the paper. T-Stochastic Neighborhood Embedding (t-SNE) was used for visualization of the data. The number of PCs used to calculate the embedding was the same as those used for the clustering. Tumor and normal epithelial cells were isolated from the whole dataset of CRC primary tissues and were analyzed using the same technical approach reported above. The number of HVG used was 951, the number of neighbors was 15 and the number of principal components was 20.

**Scoring cells of primary CRCs using signature gene sets**. Gene signature scores were calculated (the equation is shown below) given a cell by gene expression matrix and a geneset **G**. For each cell **C** in the matrix, the fraction of genes from **G** that are expressed (expression levels > 0) is computed (2). Similarly, an expression score for each cell **C** is evaluated by summing up the expression levels of genes from **G** and dividing by the total sum of gene expression levels for all genes in the same cell (3). The two scores are then multiplied together to yield a combined score for each cell **C** (4) and the reciprocal of the negative logarithm of the combined score is computed. Combined scores were created for a stemness gene set

(Supplementary Data 10) and the gene set related to the 46 pan-cancer enhancers (Supplementary Data 8).

Given a cell $C$ defined as a vector of gene expression values $[g_i, \dots, g_C]$ and a geneset $G = \{g_1, \dots, g_G\}$ a coexpression score is calculated as:

$$c\_score = \frac{\sum_{g \in G}[Cg > 0]}{|G|} \qquad (2)$$

and an expression score is defined as:

$$e\_score = \frac{\sum_{g \in G}Cg}{\sum C} \qquad (3)$$

The two scores are then combined to yield a combined score:

$$combined\_score = c\_score * e\_score \qquad (4)$$

**Statistics and reproducibility**. Statistical analyses were performed in R (v3.5.1) and graphing in Illustrator (v25.0). Statistical significance was tested with a two-sided Wilcoxon rank sum test for gene expression comparisons, and one-sided Mann–Whitney $U$ or Wilcoxon signed rank exact tests for organoid cell viability. $P$-values are reported in each figure and legend. Results in Fig. 4b and Supplementary Fig. 5f are representative of three independent repeats and in Fig. 4i for three and six independent organoid lines for normal or tumoral PDOs, respectively.

**Reporting summary**. Further information on experimental design is available in the Nature Research Reporting Summary linked to this paper.

## Data availability

The RNA-seq data used in this study are available in the European Nucleotide Archive (ENA) database under accession code E-MTAB-8448. The ChIP-seq data generated in this study have been deposited in the ENA database under accession code E-MTAB-8416. The epigenomic (ChIP-seq on histone marks and ChromHMM tracks) and transcriptomic (RNA-seq) data for all the PDOs analyzed in this study are publicly available at the *HePIC* web browser at https://bioinformatics.ifom.eu/hepic/. Publicly available datasets used in this study can be found in the Gene Expression Omnibus (GEO) database under accession codes GSE77737 (colon; H3K27ac ChIP-seq), GSE132465 (colorectal cancer; single-cell RNA-seq), GSE51776 (gastric cancer, H3K27ac ChIP-seq), GSE114737 (breast and endometrial cancer; H3K27ac ChIP-seq), GSE74230 (osteosarcoma; H3K27ac ChIP-seq), GSE101065 (uterus; H3K27ac ChIP-seq), GSE142924 (uterus; H3K27ac ChIP-seq), GSE16256 (liver, adrenal gland and pancreas; H3K27ac ChIP-seq), GSE96504 (liver; H3K27ac ChIP-seq), GSE101019 (adrenal gland; H3K27ac ChIP-seq), GSE95966 (adrenal gland; H3K27ac ChIP-seq), GSE101258 (thyroid gland; H3K27ac ChIP-seq), GSE96258 (thyroid gland; H3K27ac ChIP-seq), GSE95981 (thyroid gland; H3K27ac ChIP-seq), GSE142995 (thyroid gland; H3K27ac ChIP-seq), GSE101269 (pancreas; H3K27ac ChIP-seq), GSE101231 (pancreas; H3K27ac ChIP-seq), GSE142968 (pancreas; H3K27ac ChIP-seq), GSE96212 (pancreas; H3K27ac ChIP-seq). The remaining data are available within the Article, Supplementary Information or available from the authors upon request. Source data are provided with this paper.

## Code availability

The code used in this study has been deposited on GitHub (https://github.com/paganilab/DellaChiara_et_al_2021) and linked to the Zenodo Digital Repository (https://doi.org/10.5281/zenodo.4588460)[70].

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

## Acknowledgements

We would like to thank D. Parazzoli, M. Garrè, F. Casagrande, and E. Martini at IFOM imaging facility for technical assistance; D. Pasini and M. Zanotti for initial support with organoids; M. Miozzo for helping in clinical classification of primary tumor samples; M. Fassan, V. Guzzardo and D. Di Biagio for CRC samples and in situ hybridization. This work was supported by grants from the Italian Ministry of Education, University and Research (MIUR), and the National Council of Research of Italy (CNR) under the EPIGEN Flagship project to S.B., M.P., S.P., and G.T.; the Fondazione AIRC grant IG2016-ID. 18575 and European Research Council (ERC) CoG grant n° 617978 to M.P.; the Fondazione AIRC under 5 per Mille 2019-ID. 22759 program to S.B., M.P., and S.P.; the European Research Council (ERC) AdG grant n° 670126 to S.P. HOMIC - Human Organoid Models Integrative Center is supported by the "Fondazione Romeo ed Enrica Invernizzi" and University of Milan. M. Fakiola was supported by Fondazione Umberto Veronesi.

## Author contributions

G.D., F.G., M. Fakiola, and C.G. designed and performed the main experiments, analyzed the data, performed the majority of bioinformatics analyses, and contributed to the preparation of the manuscript. C.D. and G. Moreni. contributed to the establishment and maintenance of organoid cultures. L.A. performed in situ hybridization experiments. R.J.P.B., L.D., V.R., F. Panariello., and I.F. conducted data analysis and contributed to set up all the bioinformatics pipelines. R.B. and M.D. assisted with ChIPmentation experiments. M. Forcato., O.R., and S.B. conducted motif discovery and functional enrichment analyses. F.G., R.J.P.B., and J.C. created the web application. T.F, P.G., and M.L.S. assisted with experiments. N.Z., A.P.C., N.M.M., A.C., A.S., L.G., E.O., and S.S. coordinated clinical contributions and pathology analyses. F. Pisati. performed immunohistochemistry analysis. G.R., F.Z., M.C., A.B., G.T., C.T., G. Macino., and S.B. discussed results and commented on the manuscript. G.D., F.G., M. Fakiola., C.G., S.P., and M.P. wrote the manuscript. S.P. and M.P. designed the study and supervised the research. All authors discussed and interpreted the results.

## Competing interests

M.P. is co-founder, member of the board of directors, and stakeholder of the company CheckmAb s.r.l. The remaining authors declare no competing interests.

## Additional information

[1]IFOM, the FIRC Institute of Molecular Oncology, Milan, Italy. [2]Istituto Nazionale Genetica Molecolare INGM 'Romeo ed Enrica Invernizzi', Milan, Italy. [3]Human Organoid Models Integrative Center HOMIC, University of Milan, Milan, Italy. [4]Department of Medical Biotechnology and Translational Medicine, Università degli Studi di Milano, Milan, Italy. [5]Department of Clinical Sciences and Community Health, Università degli Studi di Milano, Milan, Italy. [6]Department of Molecular Medicine, University of Padua, Padua, Italy. [7]Department of Life Sciences, University of Modena and Reggio Emilia, Modena, Italy. [8]Candiolo Cancer Institute, FPO - IRCCS, Candiolo (TO), Italy. [9]Department of Oncology, University of Torino, Candiolo (TO), Italy. [10]Department of Pathology, San Gerardo Hospital, Monza, Italy. [11]UO Chirurgia Epatobiliopancreatica e Digestiva Ospedale San Paolo, Milan, Italy. [12]Niguarda Cancer Center, Grande Ospedale Metropolitano Niguarda, Milan, Italy. [13]Department of Oncology and Hemato-Oncology, University of Milan, Milan, Italy. [14]Department of Experimental Oncology, European Institute of Oncology, IRCCS, Milan, Italy. [15]School of Medicine and Surgery, Milano-Bicocca University, and Department of Surgery, San Gerardo Hospital, Monza, Italy. [16]Department of Health Sciences, Università degli Studi di Milano, Milan, Italy. [17]Histopathology Unit, Cogentech, Milan, Italy. [18]Tumor Immunology Unit, University of Palermo, Palermo, Italy. [19]Tumor and Microenvironment Histopathology Unit, IFOM, FIRC Institute of Molecular Oncology, Milan, Italy. [20]Department of Cellular Biotechnologies and Hematology, La Sapienza University of Rome, Rome, Italy. [21]Present address: Department of Medical Microbiology, Laboratory of Clinical Virology, Amsterdam University Medical Center, University of Amsterdam, AZ Amsterdam, the Netherlands. [22]Present address: Technology Center for Genomics and Bioinformatics, Department of Pathology and Laboratory Medicine, University of California, Los Angeles, CA, USA. [23]Present address: Telethon Institute of Genetics and Medicine TIGEM, Pozzuoli, Italy. [24]These authors contributed equally: Giulia Della Chiara, Federica Gervasoni, Michaela Fakiola, Chiara Godano. ✉email: piccolo@bio.unipd.it; massimiliano.pagani@ifom.eu

