## [Peer Review File · Nature Communications]

REVIEWER COMMENTS

Reviewer #1 (Remarks to the Author): Expert in CRC genomics

The main contribution of this paper could be a resource, library of epigenetic changes in Colon tumor PDO

There are many very strong statements about the utility of this resource:

'This de novo chromatin state characterization provided a rich reference resource 4 (available at <http://hepic.homic.eu>) of regulatory elements (promoters and enhancers), elongating and repressed genomic regions defined by multiple epigenetic features, enabling a comprehensive interrogation of the colon cancer epigenomic landscape.'

Correct, within the setting of 10 patient PDO

The extrapolations /correlations to other datasets are not necessarily foolproof or evidence of the relevance of the 10 pt PDO set.

For example:ATAC seq on PDO and tumor single cell would have been stronger, to bridge all the different sampels and understand culture effects versus in vivo biology

-The first results section is just a transcriptional comparison of tumor PDO, the original tumor and normal tissue. That PDO recapitulate largely transcriptional profiles of the tumors they derive from is known and not novel. There was 84% overlap in gene expression between tumor and PDO. The non overlapping expression is of relevance to discuss and how this impact conclusions of the paper and the use of the PDO models

-Next they have chipseq data for the PDo, bioinformatically analyzed with conclusions on enhancers states . these conclusions (after bioinformatical modeling) are compared to TCGA ATC seq data and they conclude congruent information, therefore validating the model of PDO for such studies. Again the non congruent information and the DPO specific states, overly homogenous are of critical relevance

-Recent literature highlights again that a lot of the regulation of colon crypt differentiation can be posttranslation, <https://www.nature.com/articles/s42003-020-01181-z.pdf>, questioning the relevance of using the chomatin to describe differentiation states.

-

-This could also be in line with their surprising homogeneity of chromatin states across patients, the dynamic regulation may be downstream (an/or the pDO models are too 'homogenized' by the PDO culture conditions that specifically use all the differentiating agents of which the in vivo biological effects are queried.....)

-next section on 'we sought to determine which of the genes annotated to gained enhancers were upregulated in the majority of PDOs compared to normal tissues '. Comparing the PDO in culture gene expression with normal non cultured colon tissue has the risk of picking up culture driven features

-the YAP /TAZ conclusion for CRC and pan cancer are bioinformatically derived but not experimentally. The bioinformatics approach is logical and conclusion interesting, but absolutely in need of orthogonal validation

-the single cell data if I understand correctly is derived from 1 patient...many, many conclusions are linked to this data, based on clustering and trajectories which can only be unstable at this sample size...

Reviewer #2 (Remarks to the Author): Expert in gastric organoids and tumorigenesis

This is an interesting manuscript, revealing the common core enhancers among colorectal cancers that are under the regulation of YAP/TAZ. The use of cancer organoids for the detailed epigenetic study is indeed highly recommended as we can get high-quality data with greater resolution from highly pure cancer materials without any contamination from stromal and immune cells. I recommend this manuscript for publication after the authors address the comments below:

1. First of all, the number of cancer organoid lines analyzed in this study is rather small to make a strong conclusion as there could always be a selection bias with a small number of samples. In Fig 3c, the authors pointed out that a significant percentage of gained enhancers is shared by different PDOs, especially 20% is shared by 8 patients. Which are the PDOs that are not sharing this 20%? Are they belonging to a specific subtype in Fig 1b?
2. In Fig 4b, the immunostaining of YAP/TAZ is very poor and not acceptable. The upregulation of YAP/TAZ expression is clear among the selected PDOs in Fig 4a. However, it seems necessary to check the level of YAP/TAZ expression in a larger gene expression dataset of colon cancer in order to generalize the authors' observation.
3. For the generalization of the authors' claim, their analysis in Fig 5a,b seems very important – the most convincing part of the manuscript. However, their analysis of YAP/TAZ downstream genes doesn't look so convincing as there are only a few examples. It is just difficult to accept that those few observations are representative features of all CRCs. The authors shall try to identify well-conserved YAP/TAZ-controlled gene sets that clearly show common features e.g. commonly over-expressed across many CRC samples. The more, the better but at the moment those selected pieces of evidence do not seem enough.
4. Lastly, scRNAseq analysis is very disappointing. Instead of primary cancer materials, the authors should have analyzed their normal and cancer organoids. Recently, several reports have shown the existence of fetal cell types in the mouse organoid culture that are under the control of YAP/TAZ signaling. The authors should try to find out similar fetal-like cell types in the cancer organoids. If the YAP/TAZ enhancers are fully operative in CRC PDOs, it is expected to see more fetal-like progenitors in cancer organoids compared to in normal organoids. In my opinion, the current scRNAseq data can be removed from the manuscript.

Reviewer #3 (Remarks to the Author): Expert in epigenetics and chromatin organization

Epigenetic landscape of human colorectal cancer unveils an aberrant core of pan-cancer enhancers orchestrated by YAP/TAZ by Giulia Della Chiara et al,

In this paper the authors exploit a library of patient-derived organoids to identify a tumor-specific deregulated enhancerome that is cancer cell intrinsic and independent of interpatient heterogeneity. They find that the transcriptional coactivators YAP/TAZ bind gained enhancers which harbor active chromatin profiles as a pan-cancer rewiring of importance for tumor maintenance. They further propose that a YAP/TAZ fueled enhancer reprogramming represents a common feature of the cancer cell state offering avenues for development of new therapeutics.

First, they validated their PDO library of different CRC subtype using transcriptomic data and histopathological features. They went on and performed a multifactorial analysis by ChIP-seq for H3K4me3, H3K27Ac, H3K4me1, H3K36me3 and H3K27me3 on all PDOs and found

concordant distribution across PDO from different patients. By exploiting machine learning approaches using ChromHMM, they defined two promoter states, one weak and two active enhancer states as well as an elongation, a repressed and a quiescent state. They compared these data with ATACseq data sets from the Cancer Genome Atlas. Not only they concluded that their PDOs preserved the regulatory network of primary tumors but further used their data as an atlas to interpret the ATAC-seq identified opened chromatin regions. Thereby, they compile a reference resource available at <http://hepic.homic.eu>. They defined then an enhancerome of human CRC with about 33K enhancers that distinguish PDO from normal colon tissue and examined their activation state and found a gain of 2,419 enhancers compared to normal tissue. They went on and sought for the concordant tumor-enriched enhancers in the PDOs. Remarkably, independently of their original molecular and histological features, about 20 % were shared by at least 5 patients. They further identified by motif discovery an enrichment in the TEA domain family member in these gained enhancers suggesting a role for their coactivators, the YAP/TAZ regulators in partnership with AP-1. Both YAP and TAZ proved to be upregulated and hyperactive in CRC. By ChIPseq they identified an enrichment in TAZ and YAP at sites where the TEAD family motif was most enriched.

Next, they used Single cell RNA seq in a primary CRC tissue to access to deregulation at a single cell level and they could identify clusters of undefined cell population revealing intra tumor heterogeneity with 5 types of malignant clusters. They defined a score of the cancer regulatory blueprint (CRB score) highly enriched in malignant clusters irrespective of the intra tumor genetic and transcriptional heterogeneity and that goes beyond the acquisition of stemness. They suggest that aberrant YAP/TAZ activation is required for both tumorigenesis and maintenance of the cancer cell state.

This work is an interesting piece providing very useful and novel resource material that the community will be able to exploit further. The concept of a core cellular defect across the heterogeneity of samples from patients is surely an important finding that will influence our current thinking. At this point in time, the data analysis provides strong correlations, which should be carefully checked for statistics and computational analysis. Should this part be satisfactorily examined, the one missing aspect is an experimental approach that would bring this work beyond correlation to causality. This could perhaps be addressed by interfering with the YAP/TAZ pathway using their PDO library and see whether they affect the cancer cell state maintenance, and/ or use a control organoid and alter the YAP/TAZ pathway to show that tumorigenesis is proceeding. While this may represent significant additional work, it is surely important to remain cautious in the interpretation of the data.

Another point is to try and be cautious in the use of the term epigenetics or epigenomics. Here, it seems that they are examining mostly epigenomic features. A careful editing of the manuscript will ensure to be accurate in the wording across the whole document.

In conclusion, in my opinion, this manuscript does represent an interesting piece for publication in Nature Communications and with some revisions could be accepted.

Reviewer #4 (Remarks to the Author): Expert in single-cell RNA-seq, computational biology, and statistics

Della Chiara et al use data generated from organoids and from primary tissue to transcriptionally and epigenetically characterize human colorectal cancer.

To do so, they analyze in particular a single-cell RNA-seq dataset from a primary cancer tissue. During the analysis, they could separate healthy from tumour cells and transcriptionally characterize the latter.

Overall this part of the analysis is well executed and the conclusions drawn are justified. There are a few points though that would need clarification:

1. Calling CNV from scRNAseq data (especially those generated with the 10x protocol, which are particularly sparse) is not robust, and can be confounded by differences in gene expression patterns. In my opinion, the discussion about CNV and the existence of two clones should be more cautious, and some robustness analysis should be done. For instance, what cell population was used as a reference to call the CNV? Do the results change if different cell populations are used as reference (eg, different cell types in the data)?
2. Page 14, line 20: The authors should be more specific about what constitutes the "clear deviation from normal epithelium differentiation" observed in cancer cells
3. Supp Fig 6e: A better visualization could be used here. For instance, stacked barplots showing the fraction of cells in G1,S,G2M in each cluster.
4. Page 25 line 10: how was transcript variability measured in the bulk RNAseq data?
5. Page 25 line 11: what are the immune-related genes that were removed?
6. Page 25 line 12: how were pvalues adjusted?
7. For the scRNAseq data, it would be useful to show the statistics of the QC metrics of the good quality cells (eg, number of UMIs, number of genes detected, % of mit genes, etc)
8. Page 35, lines 10-17: this description would be much clearer if the authors wrote an equation for the score they compute

Reviewer #5 (Remarks to the Author): Expert in bioinformatics and epigenetics

The results presented by the authors are very interesting, as well as their integrated approach to the identification of a core set of pan-cancer enhancers. The manuscript provides a valuable example of how genome-wide analyses in patient-derived organoids (PDOs) can be successfully harnessed to define common epigenetic features across heterogeneous cancers and cell subpopulations.

The role of TAZ, however, should be evaluated more carefully. A number of potential issues need to be addressed to support the conclusion that TAZ binding is key. Further analyses would also help support this point. A list of suggestions to strengthen the manuscript are listed below. The overall evaluation remains nonetheless positive, and the approach presented by the authors is very promising.

1) Fig. 4g: this comparison supports the hypothesis that TAZ binding is a common feature of CRC enhancers across different PDOs, but the methodology is a bit unclear and there may be some potential caveats that should be addressed more carefully.

The first caveat concerns the randomization and testing procedure. Was bedtools shuffle run with default parameters? The authors state that they used bedtools shuffle "to generate 1000 shuffled tracks", which suggests that 1000 sets were generated by randomly permuting the 2419 enhancer locations (i.e. 1000 BED files were produced from the original file). If this is the case, how was the percentage in Figure 4c computed and the Fisher's test performed?

If 1000 sets with 2419 random locations were generated, the percentage of random locations overlapping a TAZ peak should be computed (separately) for each of the 1000

permutations. This would enable to compare the observed fraction of TAZ-bound GEs to the distribution in the 1000 random sets, and thus compute an empirical p-value i.e. # of times the % of TAZ -bound locations in a random set is as extreme as the observed % of TAZ-bound enhancers, divided by 1000. It's not clear instead how a p-value would be computed via a Fisher's test, since summing over 2419 x 1000 locations would have the effect of artificially inflating the column total for the control set (and thus decrease the p-value). The authors should better describe their approach, and the distribution of the % overlap over 1000 permutations should be taken into account.

Another potential issue with this comparison is that the size distribution of the enhancers is likely to change with the level of conservation among PDOs. It's true that the distribution of the shuffled set would match the one of the original set of gained enhancers (GEs) but this does not necessarily apply to the subset of GEs shared by e.g. 8/10 samples, which instead might be larger. The probability that an enhancer found in one will overlap with the enhancer found in another sample is, arguably, higher for larger enhancers than shorter ones. The same would similarly apply to the probability of an enhancer overlapping a TAZ peak. Control sets should be thus generated for all GE subsets in Fig. 4c e.g. the % of TAZ bound enhancers shared by 8/10 PDOs should be compared to the % in a random set where the locations of GEs shared by 8/10 PDOs are shuffled (rather than a control set where all GE locations are shuffled).

Alternatively, the authors could compare the size distribution of each subset (i.e. GEs, GEs shared by 5/10 samples, GEs shared by 8/10 samples) and make sure that enhancer size does not increase with the level of conservation across PDOs.

Further analyses would also help strengthen the conclusion. The authors could, for instance, compare TAZ signal intensity at enhancers overlapping in 1/10, 5/10, 8/10 PDOs (heatmap and average signal) or TAZ binding in cancer vs normal tissue.

2) Fig. 5: the results are very interesting, but to support the role of TAZ as a key factor it would be important to compare the set of TAZ-regulated enhancers to a control set where TAZ is not found (e.g. the GEs shared by 8/10 PDOs that do not overlap TAZ peaks, or GEs with a low average TAZ signal relative to input vs GEs with a higher average signal). This would allow to a) assess if TAZ binding is a defining feature of CRC and pan-cancer enhancers, b) evaluate the number of pan-cancer enhancers that can be identified independently of TAZ, and c) check if H3K27ac differences relative to normal tissues are e.g. larger for TAZ-regulated enhancers.

3) Fig. 6: in line with the previous comment, it would be informative to compute an expression signature including genes that are associated with TAZ-independent enhancers, and compare cell-to-cell variation for TAZ-independent genes to the CRB signature from TAZ-regulated pan-cancer genes. It would be similarly interesting to compare signatures from the pan-cancer set to genes associated with the CRC-specific set of TAZ-regulated enhancers, and assess differences among clones and subpopulations of cells.

4) Although not crucial, some parts of the methodology are explained very briefly and a more detailed description would improve the reproducibility of the results. For the differential expression analysis: are samples paired or unpaired? What group-level comparison and test was performed? For enhancer-promoter associations: what is the rationale for using different approaches? Is distance or overlap the criterion when Hi-C data is not used?

Response to Reviewers' Comments:
Epigenetic landscape of human colorectal cancer unveils
an aberrant core of pan-cancer enhancers orchestrated by YAP/TAZ

We thank the Reviewers and the Editor for their careful assessment and insightful comments on the original version of our manuscript. Since receiving the reviews we have undertaken additional experimental and analytical work, refining the manuscript and strengthening its conclusions. We hope these changes address the various issues raised with the previous version.

In what follows, we first explain the major changes and additional work since the original submission, and then respond in detail to each of the Reviewers' specific comments.

Overall Changes

The key/major changes to the revised manuscript are as follows:

- **Experiments to validate the role of YAP/TAZ in cancer maintenance.** We treated patient-derived organoids with verteporfin (VP), that inhibits YAP/TAZ by disrupting their interaction with the transcriptional co-activator TEAD (Imajo et al. 2014 *Nature Cell Biology*; Cebola et al. 2015 *Nature Cell Biology*). YAP/TAZ inhibition following VP-treatment had a significant impact in tumor compared to normal organoids leading to growth suppression and extensive cell death. These experiments confirm the central role of YAP/TAZ as epigenetic regulators of cancer cells.
- **New analysis of single-cell RNA-seq data of 23 CRC patients**, confirming the tumor-specific nature of the YAP/TAZ-regulated pan-cancer enhancers. We found that the expression signature of genes annotated to the pan-cancer enhancers is highly enriched in malignant epithelial cells but largely absent from both normal epithelial and other stromal/immune cells. In addition, this signature defines transcriptional states that are not characterized by an enrichment of stem-like cell marker, suggesting that the YAP/TAZ-mediated epigenetic dysregulation is a feature of cancer, not related to stemness *per se*. Thus, the evidence for the central role of YAP/TAZ in defining the cancer cell state across heterogenous CRCs has become considerably stronger in the revised version of the paper.
- **Validation of the expression of YAP, TAZ and YAP/TAZ-regulated genes in the colon adenocarcinoma dataset of The Cancer Genome Atlas (TCGA).** By assessing RNA-seq data for 500 colon cancer patients, we confirmed that YAP, TAZ, and the genes associated to the YAP/TAZ-bound active enhancers have significantly elevated expression in tumor compared to normal tissue samples.
- **New permutation analyses to assess TAZ enrichment in gained enhancers**, confirming that TAZ occupancy increases with enhancer conservation.

- **Comparison analyses between TAZ and non-TAZ-regulated enhancers confirms the key role of TAZ in cancer.** In agreement with their differential activation in PDOs, the TAZ-independent conserved CRC-gained enhancers are more active in primary tumors than normal samples. However, both the H3K27ac differences relative to normal tissue, and the expression specificity of enhancer genes in malignant cells are lower for TAZ-independent compared to the TAZ-regulated gained enhancers.
- **We provided additional information in the Methods and Results to describe our methodology and findings in more detail.**

Response to Reviewers' Comments

In what follows the reviewers' comments appear in blue and in italics, with our responses in ordinary (black) font.

Response to Reviewer 1

Reviewer #1 (Remarks to the Author): Expert in CRC genomics

The main contribution of this paper could be a resource, library of epigenetic changes in Colon tumor PDO.

We thank the Reviewer for recognizing the value of our epigenetic resource and for their useful feedback which prompted us to expand our manuscript with further experiments and computational analyses.

There are many very strong statements about the utility of this resource: 'This de novo chromatin state characterization provided a rich reference resource (available at <http://hepic.homic.eu>) of regulatory elements (promoters and enhancers), elongating and repressed genomic regions defined by multiple epigenetic features, enabling a comprehensive interrogation of the colon cancer epigenomic landscape.' Correct, within the setting of 10 patient PDO. The extrapolations /correlations to other datasets are not necessarily foolproof or evidence of the relevance of the 10 pt PDO set. For example: ATAC seq on PDO and tumor single cell would have been stronger, to bridge all the different samples and understand culture effects versus in vivo biology.

We appreciate the Reviewer's comment which allows us to better explain the rationale of our strategy for deciphering the shared aberrant enhancerome of tumor cells and its major regulators. Performing a systematic characterization of chromatin states (based on multiple histone marks) in tumor cells is not trivial and can be particularly challenging when using primary tumors due to i) the limited number of cells in primary tumor samples, and ii) the cellular heterogeneity of the tumor microenvironment which cannot capture the cancer cell-intrinsic profiles. With this in mind, we leveraged primary tumor-derived tumor organoids (PDOs) by selecting a balanced library of 10 PDO lines representative of CRC molecular heterogeneity, which enabled us to define

the chromatin states of pure cancer cells, uncoupled from stromal/immune-cell influences. The *de novo* reconstruction of chromatin states based on ChIP-seq data for 5 histone marks also allowed us to identify tumor-specific active enhancers with high accuracy, distinguishing them from broadly-defined regulatory elements based on open chromatin regions or individual histone marks. The above two objectives, crucial for deciphering the common epigenetic blueprints of tumor cells, cannot be achieved efficiently using bulk chromatin accessibility data on primary tumors, *e.g.* TCGA ATC-seq datasets. Having identified with confidence the aberrant active enhancerome and its major regulators, we were then able to perform a targeted interrogation of primary tumor ATAC-seq data to highlight a pan-cancer core of enhancers and subsequently validate our findings using multi-omic data for independent datasets of primary tumors. This strategy also ensures that our conclusions are not influenced by culture driven conditions. In the revised version of the manuscript, the validation of our pan-cancer enhancers includes the interrogation of not only histone mark profiles but also single-cell gene expression data for a larger number of diverse primary tumors as suggested by the Reviewer. We describe these analyses in detail below:

First, we used ATAC-seq data from 400 TCGA samples to show that the YAP/TAZ-bound CRC-enhancerome was accessible across 38 colon adenocarcinoma (COAD) samples (**Fig. 5a**). A subset of these enhancers (pan-cancer core) was also shared among diverse tumor types.

Second, by investigating H3K27ac data for 43 primary tumor and normal tissue samples, we confirmed that the pan-cancer enhancers display significantly higher H3K27ac levels in primary tumors compared to normal tissues (**Fig. 5b**).

We have now improved our manuscript by interrogating single-cell RNA sequencing (scRNA-seq) data for 23 heterogeneous CRC samples (**Fig. 6 and Supplementary Fig. 6b-d**). This analysis showed that the expression of genes associated to the YAP/TAZ-regulated pan-cancer enhancers (cancer regulatory blueprint) is indeed enriched in malignant epithelial cells compared to normal epithelial or other cell types.

Collectively, multi-omic datasets consisting of a large number of primary tumor samples concur with our findings in PDOs that a YAP/TAZ-regulated core of enhancers represents a conserved feature of epigenetic deregulation in different tumor types.

-The first results section is just a transcriptional comparison of tumor PDO, the original tumor and normal tissue. That PDO recapitulate largely transcriptional profiles of the tumors they derive from is known and not novel. There was 84% overlap in gene expression between tumor and PDO. The non-overlapping expression is of relevance to discuss and how this impact conclusions of the paper and the use of the PDO models.

We thank the reviewer for raising this point, which we also considered in our study. The transcriptomic differences between primary tumors and PDOs are primarily due to the lack of a stromal component in PDOs. In the original version of the manuscript, we showed (**Supplementary Fig. 1c**) that the genes expressed specifically in primary tumors (n = 3,412) are enriched in a gene signature of stromal and immune cells, comprising endothelial cells, fibroblasts and leukocytes. This is consistent with previous reports (Isella et al., 2015, Calon et al., 2015, Fujii et al., 2016), showing a prominent stromal component in tumor tissues compared to PDOs. Thus, our findings indicate that PDOs retain the CRC gene signature but are deprived of stromal contamination, providing an advantage in deciphering the CRC molecular profiles inherent to cancer cells.

As the Reviewer noted, we also identified 534 genes that were not present in primary tumors. However, upon further investigation we found that these were not widely expressed across all PDOs, suggesting that their expression is less likely to be driven by culture conditions. Importantly, these genes are neither upregulated in PDOs compared to normal tissues nor associated to the 195 YAP/TAZ-bound conserved enhancers (**Fig. 5a**), thus they do not compromise the use of PDOs or the conclusions of the paper.

-Next they have chipseq data for the PDO, bioinformatically analyzed with conclusions on enhancers states. these conclusions (after bioinformatical modeling) are compared to TCGA ATC seq data and they conclude congruent information, therefore validating the model of PDO for such studies. Again the non-congruent information and the DPO specific states, overly homogenous are of critical relevance.

The aim of our work was to investigate the shared features of epigenetic deregulation in cancer (**Fig. 3c**) using a PDO library that represents the molecular heterogeneity of CRC (**Fig. 1b**). As the Reviewer noted, we sought to solidify our findings by comparing these conserved cancer-cell intrinsic epigenetic features to primary tumor ATAC-seq and H3K27ac data (**Fig. 2d, 5a, b**). We note that whilst the non-congruent information that highlights private regulatory elements and might relate to CRC clinical heterogeneity would be of interest, it is beyond the scope of the manuscript, and will have to be addressed in future studies. It also does not influence the conclusions of our study as we explain in our response to the Reviewer's previous comment. We apologize for the lack of clarity regarding the homogeneity of the PDO chromatin states, which we now explain below.

-Recent literature highlights again that a lot of the regulation of colon crypt differentiation can be posttranslation, <https://www.nature.com/articles/s42003-020-01181-z.pdf>, questioning the relevance of using the chromatin to describe differentiation states.

We thank the Reviewer for this comment and agree that post-transcriptional processes, such as alternative mRNA splicing and polyadenylation, as well as post-translational modifications constitute additional layers of regulation that impact colon crypt homeostasis. Although our epigenetic study focuses on the importance of enhancer usage in cancer cell reprogramming, we now cite in the Discussion of the manuscript the study by Habowski et al (Communications Biology 2020) to acknowledge the role of post-transcriptional processes.

-This could also be in line with their surprising homogeneity of chromatin states across patients, the dynamic regulation may be downstream (an/or the pDO models are too 'homogenized' by the PDO culture conditions that specifically use all the differentiating agents of which the in vivo biological effects are queried.....).

We appreciate the Reviewer's comment and regret the lack of clarity in presenting data related to chromatin state analyses (**Fig. 2b and Supplementary Fig. 2a**). Using principal component and hierarchical clustering analyses we aimed to show the global deposition of histone marks and not suggest that PDOs are epigenetically homogenous. Based on the most highly variable regions, the HC and PC analyses show a clear division between the histone marks that represent the major types of chromatin regulation, namely activation, elongation, and repression; with samples within each of these histone marks being clustered together. However, this does not preclude the presence of heterogeneity across PDO samples for a specific histone mark. In fact, in the case of active enhancers, defined by the co-presence of H3K27ac and H3K4me1, only 20% of the gained enhancers are highly conserved (present in 8/10 PDOs; **Fig 3c**). This heterogeneity is further reflected transcriptionally (**Fig. 1d**); PDOs are not grouped together but display the same spatial distribution along PC2 with their primary tumors. We now amended the description of the chromatin state analyses in the revised version of the paper to make these points clearer.

-next section on 'we sought to determine which of the genes annotated to gained enhancers were upregulated in the majority of PDOs compared to normal tissues '. Comparing the PDO in culture gene expression with normal non cultured colon tissue has the risk of picking up culture driven features.

We thank the Reviewer for this important comment. The genes annotated to gained enhancers (G.E.) that are differentially upregulated in PDOs also show significantly increased expression in primary tumors compared to normal tissues. We included a new graph (**Supplementary Fig. 4d**) that shows their level of expression in PDOs, primary tumors and normal tissues. To validate our findings, we repeated the functional enrichment analysis using G.E. genes that are upregulated in both PDOs and primary tumors compared to the normal tissues based on differential gene expression analyses (using DESeq). Notably, we again identified the Hippo Signaling Pathway among the most significantly enriched pathways. This increases the confidence in our original finding and diminishes the possibility of any influences caused by culture-driven conditions.

-the YAP /TAZ conclusion for CRC and pan cancer are bioinformatically derived but not experimentally. The bioinformatics approach is logical and conclusion interesting, but absolutely in need of orthogonal validation.

We agree with the Reviewer's comment, which prompted us to assess the role of YAP/TAZ in tumor maintenance. To this end, we treated fully-formed tumor and normal organoids with verteporfin to disrupt the interaction of YAP/TAZ with their partner transcriptional factor TEAD. Following VP-treatment, the growth and cell viability of tumor organoids was significantly affected compared to normal organoids (**Fig. 4i-j and Supplementary Fig. 5g**), strongly supporting a direct role for YAP/TAZ on the epigenetic regulation of cancer cells.

-the single cell data if I understand correctly is derived from 1 patient...many, many conclusions are linked to this data, based on clustering and trajectories which can only be unstable at this sample size...

We agree with the Reviewer that the scRNA-seq observations made in the original version of the manuscript should be validated across a larger number of CRC samples. In the revised manuscript, we performed a detailed analysis of scRNA-seq data for a dataset of 23 primary CRC patients and toned down the claims based on clustering and trajectories made in the previous version. The new analysis shows that the distribution of the cancer regulatory blueprint is enriched in malignant epithelial cells compared to both normal epithelial and non-epithelial cell populations (**Fig. 6 and Supplementary Fig. 6b-d**) confirming our conclusion for the role of YAP/TAZ in orchestrating a conserved tumor-specific core of active enhancers.

Response to Reviewer 2

Reviewer #2 (Remarks to the Author): Expert in intestinal organoids and tumorigenesis

This is an interesting manuscript, revealing the common core enhancers among colorectal cancers that are under the regulation of YAP/TAZ. The use of cancer organoids for the detailed epigenetic study is indeed highly recommended as we can get high-quality data with greater resolution from highly pure cancer materials without any contamination from stoma and immune cells. I recommend this manuscript for publication after the authors address the comments below:

We thank the Reviewer for their appreciation in our work and insightful comments, which we addressed with additional experiments and analyses which helped us to improve the manuscript.

1. First of all, the number of cancer organoid lines analyzed in this study is rather small to make a strong conclusion as there could always be a selection bias with a small number of samples.

We thank the Reviewer and agree that a small number of samples can encompass a selection bias when investigating common features of cancer. For this reason, we were careful to select 10 PDOs (from our biobank), derived from primary tumors that are representative of the genetic and molecular heterogeneity in CRC. In addition, the use of PDOs was central for identifying the tumor-associated YAP/TAZ-driven enhancerome for two main reasons. By exploiting tumor organoids, we were able to i) dissect the cancer cell-intrinsic epigenetic alterations devoid of the influence of stromal and immune cells, and ii) perform a systematic discovery of chromatin states based on the ChIP-seq profiles of five histone marks, which requires cell abundance not compatible with primary tumor samples. Thus, this approach allowed us to define tumor cell-associated active enhancers with high resolution in contrast to using ATAC-seq-defined open chromatin regions that are merely indicative of diverse active regulatory elements. Following the identification of a YAP/TAZ-controlled active enhancerome in PDOs, it was possible to perform a targeted interrogation of chromatin accessibility data for a much larger dataset of 400 primary tumors to reveal a core of enhancers that is shared amongst diverse cancer types. To consolidate our conclusions, we investigated multiple omic data, including H3K27ac patterns and single cell transcriptomics, to show that the pan-cancer enhancers are highly activated in primary tumors compared to normal tissues with the associated enhancer genes also being enriched in malignant epithelial cells. We provide further description of our strategy, datasets and findings in our response to Reviewer #1. Overall, these analyses strengthen the conclusions of our manuscript, confirming the presence of a conserved layer of enhancer reprogramming in cancer that is orchestrated by YAP/TAZ.

In Fig 3c, the authors pointed out that a significant percentage of gained enhancers is shared by different PDOs, especially 20% is shared by 8 patients. Which are the PDOs that are not sharing this 20%? Are they belonging to a specific subtype in Fin 1b?

We thank the Reviewer for the comment and apologize for the lack of clarity in our presentation of data. The distribution of conserved enhancers is similar across PDOs, with 80% or more of the conserved enhancers being activated in each PDO. We also did not observe any correlation between the number of shared enhancers and the molecular classification of PDOs shown in **Fig. 1b**. We now included a graph (**Supplementary Fig. 4c**) with the number of highly recurrent enhancers (8-10) in each PDO, clearly showing that all PDOs share a large fraction of conserved enhancers.

2a. In Fig 4b, the immunostaining of YAP/TAZ is very poor and not acceptable.

We thank the Reviewer for this comment and we now performed new immunohistochemistry analysis of primary tumors, normal tissues and organoids that more clearly depicts the subcellular localization of the YAP/TAZ proteins. We present the new data in the updated **Fig. 4b**.

2b. The upregulation of YAP/TAZ expression is clear among the selected PDOs in Fig 4a. However, it seems necessary to check the level of YAP/TAZ expression in a larger gene expression dataset of colon cancer in order to generalize the authors' observation.

We thank the Reviewer for this useful suggestion. Using RNA-seq data of 500 colon adenocarcinoma (COAD) samples from the TCGA dataset, we examined the levels of YAP and TAZ expression and confirmed their significant upregulation ($P < 0.0001$, Wilcoxon rank sum test) in tumor compared to normal tissue samples. We added a boxplot showing the upregulation of YAP/TAZ in COAD in the Supplementary (**Supplementary Fig. 5a**).

3. For the generalization of the authors' claim, their analysis in Fig 5a,b seems very important – the most convincing part of the manuscript. However, their analysis of YAP/TAZ downstream genes doesn't look so convincing as there are only a few examples. It is just difficult to accept that those few observations are representative features of all CRCs. The authors shall try to identify well-conserved YAP/TAZ-controlled gene sets that clearly show common features e.g. commonly over-expressed across many CRC samples. The more, the better but at the moment those selected pieces of evidence do not seem enough.

We thank the Reviewer for this excellent suggestion, which prompted us to further investigate the YAP/TAZ enhancer regions. To identify the putative target genes of the 195 conserved and YAP/TAZ-controlled enhancers we interrogated capture Hi-C data on human colon or identified the nearest gene with a ChromHMM-defined active promoter state. To capture genes that display enhanced expression, we integrated our transcriptional data and selected the target genes that are upregulated in PDOs compared to normal tissues. We subsequently determined their expression across 500 colon adenocarcinoma (COAD) samples confirming the upregulation of the vast majority of them in tumor versus normal tissues also in the TCGA dataset. We included a heatmap of the transcriptional expression of the target genes in the Supplementary (**Supplementary Fig. 6a**).

4. Lastly, scRNAseq analysis is very disappointing. Instead of primary cancer materials, the authors should have analyzed their normal and cancer organoids. Recently, several reports have shown the existence of fetal cell types in the mouse organoid culture that are under the control of YAP/TAZ signaling. The authors should try to find out similar fetal-like cell types in the cancer organoids. If the YAP/TAZ enhancers are fully operative

in CRC PDOs, it is expected to see more fetal-like progenitors in cancer organoids compared to in normal organoids. In my opinion, the current scRNAseq data can be removed from the manuscript.

In our manuscript, we aimed to tackle key biological questions regarding the epigenetic alterations that occur in tumor cells. Given the challenges of addressing these questions in *ex vivo* primary tumors, we performed a systematic epigenetic characterization leveraging the organoids model. However, to extend the relevance of our findings beyond patient-derived organoid models, it is important to validate them in a larger dataset of heterogeneous primary tumors, providing more compelling evidence for the role of YAP/TAZ in tumor-specific enhancer reprogramming than follow-up analyses in normal and tumor organoids.

We agree with the Reviewer that the scRNA-seq analysis in the previous version of the manuscript lacked confidence. We thus expanded the scRNA-seq analyses (**Fig. 6 and Supplementary Fig. 6b-d**) to investigate primary tumors and normal samples from 23 CRC patients, showing that the cancer regulatory blueprint (expression signature of genes annotated to YAP/TAZ-controlled pan-cancer enhancers) is highly active and specific for malignant epithelial cells. This analysis further confirmed the conclusions of our study.

The scRNA-seq data we added also confirms the previous observation that the cancer regulatory blueprint is not restricted to the tumor cell compartment enriched in stem markers but is widespread across different tumor cell clusters. This is in line with the diverse mechanisms of YAP/TAZ activity in cancer that reach beyond the acquisition of stemness and in support of our hypothesis that YAP/TAZ is necessary not only for tumorigenesis but also maintenance of the cancer cell state.

Response to Reviewer 3

Reviewer #3 (Remarks to the Author): Expert in epigenetics and chromatin organization

Epigenetic landscape of human colorectal cancer unveils an aberrant core of pan-cancer enhancers orchestrated by YAP/TAZ by Giulia Della Chiara et al. In this paper the authors exploit a library of patient-derived organoids to identify a tumor-specific deregulated enhancerome that is cancer cell intrinsic and independent of interpatient heterogeneity. They find that the transcriptional coactivators YAP/TAZ bind gained enhancers which harbor active chromatin profiles as a pan-cancer rewiring of importance for tumor maintenance. They further propose that a YAP/TAZ fueled enhancer reprogramming represents a common feature of the cancer cell state offering avenues for development of new therapeutics.

First, they validated their PDO library of different CRC subtype using transcriptomic data and histopathological features. They went on and performed a multifactorial analysis by ChIP-seq for H3K4me3, H3K27Ac, H3K4me1, H3K36me3 and H3K27me3 on all PDOs and found concordant distribution across PDO from different patients. By exploiting machine learning approaches using ChromHMM, they defined two

promoter states, one weak and two active enhancer states as well as an elongation, a repressed and a quiescent state. They compared these data with ATACseq data sets from the Cancer Genome Atlas. Not only they concluded that their PDOs preserved the regulatory network of primary tumors but further used their data as an atlas to interpret the ATAC-seq identified opened chromatin regions. Thereby, they compile a reference resource available at <http://hepic.homic.eu>.

They defined then an enhancerome of human CRC with about 33K enhancers that distinguish PDO from normal colon tissue and examined their activation state and found a gain of 2,419 enhancers compared to normal tissue. They went on and sought for the concordant tumor-enriched enhancers in the PDOs. Remarkably, independently of their original molecular and histological features, about 20 % were shared by at least 5 patients. They further identified by motif discovery an enrichment in the TEA domain family member in these gained enhancers suggesting a role for their coactivators, the YAP/TAZ regulators in partnership with AP-1. Both YAP and TAZ proved to be upregulated and hyperactive in CRC. By ChIPseq they identified an enrichment in TAZ and YAP at sites where the TEAD family motif was most enriched. Next, they used Single cell RNA seq in a primary CRC tissue to assess to deregulation at a single cell level and they could identify clusters of undefined cell population revealing intra tumor heterogeneity with 5 types of malignant clusters. They defined a score of the cancer regulatory blueprint (CRB score) highly enriched in malignant clusters irrespective of the intra tumor genetic and transcriptional heterogeneity and that goes beyond the acquisition of stemness. They suggest that aberrant YAP/TAZ activation is required for both tumorigenesis and maintenance of the cancer cell state.

This work is an interesting piece providing very useful and novel resource material that the community will be able to exploit further. The concept of a core cellular defect across the heterogeneity of samples from patients is surely an important finding that will influence our current thinking.

We thank the Reviewer for his enthusiasm and positive assessment of our work.

At this point in time, the data analysis provides strong correlations, which should be carefully checked for statistics and computational analysis. Should this part be satisfactorily examined, the one missing aspect is an experimental approach that would bring this work beyond correlation to causality. This could perhaps be addressed by interfering with the YAP/TAZ pathway using their PDO library and see whether they affect the cancer cell state maintenance, and/ or use a control organoid and alter the YAP/TAZ pathway to show that tumorigenesis is proceeding. While this may represent significant additional work, it is surely important to remain cautious in the interpretation of the data.

We thank the Reviewer for raising this important issue and agree that experimental verification of the role of YAP/TAZ in cancer maintenance will strengthen the conclusion of our manuscript. We now added experiments showing that the growth of tumor organoids is strongly suppressed following treatment with verteporfin, an inhibitor of YAP/TAZ-TEAD interactions (**Fig. 4i-j and Supplementary Fig. 5g**). VP treatment significantly

reduced the cell viability of tumor organoids compared to normal organoids, confirming that YAP/TAZ function is crucial for maintenance of the cancer cell state but is dispensable for normal tissue homeostasis.

Another point is to try and be cautious in the use of the term epigenetics or epigenomics. Here, it seems that they are examining mostly epigenomic features. A careful editing of the manuscript will ensure to be accurate in the wording across the whole document.

We thank the Reviewer for this suggestion. The text has been amended in the revised version of the manuscript.

In conclusion, in my opinion, this manuscript does represent an interesting piece for publication in Nature Communications and with some revisions could be accepted.

To address the issues raised by all the reviewers, we revised our manuscript expanding data analyses and including experimental data as summarized in the **Overview** above.

Response to Reviewer 4

Reviewer #4 (Remarks to the Author): Expert in single-cell RNA-seq, computational biology, and statistics

Della Chiara et al use data generated from organoids and from primary tissue to transcriptionally and epigenetically characterize human colorectal cancer. To do so, they analyze in particular a single-cell RNA-seq dataset from a primary cancer tissue. During the analysis, they could separate healthy from tumour cells and transcriptionally characterize the latter. Overall this part of the analysis is well executed and the conclusions drawn are justified. There are a few points though that would need clarification:

We thank the Reviewer for their careful evaluation and thoughtful comments. We now performed a new scRNA-seq analysis on a larger CRC dataset that has replaced the analysis in the previous version of the manuscript and has led to more precise and clear findings.

1. Calling CNV from scRNAseq data (especially those generated with the 10x protocol, which are particularly sparse) is not robust, and can be confounded by differences in gene expression patterns. In my opinion, the discussion about CNV and the existence of two clones should be more cautious, and some robustness analysis

should be done. For instance, what cell population was used as a reference to call the CNV? Do the results change if different cell populations are used as reference (eg, different cell types in the data)?

We thank the Reviewer for raising this important issue. Our new analysis of scRNA-seq data for 23 CRC patients uses the cell type designation of the original study and thus no longer includes CNV calling. **Current Fig. 6** and **Supplementary Fig. 6b-d** show the expression distribution of the genes annotated to the pan-cancer enhancers (cancer regulatory blueprint) in clinically-defined tumor and normal colon tissues of the 23 CRC patients. This analysis supports our previous finding that malignant epithelial cells are enriched for the CRB score compared not only to normal epithelial but also other non-epithelial cell populations.

2. Page 14, line 20: The authors should be more specific about what constitutes the "clear deviation from normal epithelium differentiation" observed in cancer cells.

We apologize for the confusion in our phrasing. In the previous version of the manuscript, this sentence referred to the lack of a clear transcriptional state of differentiation in tumor cells as defined by previously reported reference gene sets of differentiated normal intestinal epithelium. In new improved analysis, this sentence is not relevant and has thus been removed.

3. Supp Fig 6e: A better visualization could be used here. For instance, stacked barplots showing the fraction of cells in G1,S,G2M in each cluster.

We thank the Reviewer for this suggestion. Due to the updated scRNA-seq analysis, we no longer report the expression of cell cycle genes across clusters. Although the in-depth investigation of the scRNA-seq dataset of 23 CRC patients is already published (Lee et al. Nat Genet. 2020), we show below a stacked barplot (**Figure R1**) with the fraction of cells in G1, S, and G2M in each of the identified clusters.

Figure R1. Stacked barplots showing the fraction of cells in G1, S, and G2M in each cluster of the scRNA-seq dataset of 23 CRC patients.

4. Page 25 line 10: how was transcript variability measured in the bulk RNAseq data?

We thank the Reviewer for pointing out this omission in the description of the RNA-seq analysis workflow. Transcript variability was assessed by calculating the sample variance for each gene (using the *rowVars* function of R across all samples) and subsequently selecting the top 500 most variable genes. We now updated the **Methods** to include this information.

5. Page 25 line 11: what are the immune-related genes that were removed?

We thank the Reviewer for this comment. To perform principal component analysis, we selected the top 500 most variable genes and removed those with immune- or stroma-related terms based on gene ontology analysis. The list of genes is now included as **Supplementary Table 2**.

6. Page 25 line 12: how were p-values adjusted?

The p-values were corrected for multiple testing using the Benjamini-Hochberg method, the default method in DESeq. We now added this information in the **Methods**.

7. For the scRNAseq data, it would be useful to show the statistics of the QC metrics of the good quality cells (eg, number of UMIs, number of genes detected, % of mit genes, etc).

We agree with the Reviewer, though we no longer include in our manuscript scRNAseq data for our single primary tumor. QC metrics for the scRNAseq data of the CRC dataset are reported in the original publication but we reproduced plots of summary statistics, shown below (**Figure R2**).

Figure R2. Quality control metrics for scRNA-seq data of 23 CRC patients, depicting the distribution of mitochondrial genes (top left), ribosomal genes (top right) number of genes per cell (bottom left) and number of counts per cell (bottom right).

8. Page 35, lines 10-17: this description would be much clearer if the authors wrote an equation for the score they compute.

We agree with the reviewer and apologize for this oversight. We now included the equation for the CRB score in the Methods.

Response to Reviewer 5

Reviewer #5 (Remarks to the Author): Expert in bioinformatics and epigenetics

The results presented by the authors are very interesting, as well as their integrated approach to the identification of a core set of pan-cancer enhancers. The manuscript provides a valuable example of how genome-wide analyses in patient-derived organoids (PDOs) can be successfully harnessed to define common epigenetic features across heterogeneous cancers and cell subpopulations.

The role of TAZ, however, should be evaluated more carefully. A number of potential issues need to be addressed to support the conclusion that TAZ binding is key. Further analyses would also help support this point. A list of suggestions to strengthen the manuscript are listed below. The overall evaluation remains nonetheless positive, and the approach presented by the authors is very promising.

We thank the Reviewer for their positive assessment and helpful advice, which allowed us to strengthen our manuscript. To address these specific comments, we have expanded the data analyses and experiments in the revised manuscript, as summarized in the **Overview** above.

1) Fig. 4g: this comparison supports the hypothesis that TAZ binding is a common feature of CRC enhancers across different PDOs, but the methodology is a bit unclear and there may be some potential caveats that should be addressed more carefully.

The first caveat concerns the randomization and testing procedure. Was bedtools shuffle run with default parameters? The authors state that they used bedtools shuffle “to generate 1000 shuffled tracks”, which suggests that 1000 sets were generated by randomly permuting the 2419 enhancer locations (i.e. 1000 BED files were produced from the original file). If this is the case, how was the percentage in Figure 4c computed and the Fisher’s test performed? If 1000 sets with 2419 random locations were generated, the percentage of random locations overlapping a TAZ peak should be computed (separately) for each of the 1000 permutations. This would enable to compare the observed fraction of TAZ-bound GEs to the distribution in the 1000 random sets, and thus compute an empirical p-value i.e. # of times the % of TAZ-bound locations in a random set is as extreme as the observed % of TAZ-bound enhancers, divided by 1000. It’s not clear instead how a p-value would

be computed via a Fisher's test, since summing over 2419 x 1000 locations would have the effect of artificially inflating the column total for the control set (and thus decrease the p-value). The authors should better describe their approach, and the distribution of the % overlap over 1000 permutations should be taken into account.

Another potential issue with this comparison is that the size distribution of the enhancers is likely to change with the level of conservation among PDOs. It's true that the distribution of the shuffled set would match the one of the original set of gained enhancers (GEs) but this does not necessarily apply to the subset of GEs shared by e.g. 8/10 samples, which instead might be larger. The probability that an enhancer found in one will overlap with the enhancer found in another sample is, arguably, higher for larger enhancers than shorter ones. The same would similarly apply to the probability of an enhancer overlapping a TAZ peak. Control sets should be thus generated for all GE subsets in Fig. 4c e.g. the % of TAZ bound enhancers shared by 8/10 PDOs should be compared to the % in a random set where the locations of GEs shared by 8/10 PDOs are shuffled (rather than a control set where all GE locations are shuffled). Alternatively, the authors could compare the size distribution of each subset (i.e. GEs, GEs shared by 5/10 samples, GEs shared by 8/10 samples) and make sure that enhancer size does not increase with the level of conservation across PDOs.

We thank the Reviewer for raising this important point and apologize for any lack of clarity in our presentation. Figure 4g shows the percentage of gained enhancers within each group (x-axis) that are bound by TAZ. For instance, 40% of conserved G.E. (195 out of 486) are bound by TAZ compared to 18.4% of all G.E. (446 out of 2,419). As the Reviewer noted, to test the significance of our observation, we generated 1000 shuffled tracks using bedtools shuffle with default parameters. Each feature in the input BED file was repositioned in genomic regions of the ChromHMM-defined enhancerome ($n = 33,131$ PDO and normal tissue enhancers), excluding the genomic regions of PDO gained enhancers.

The Reviewer's comment rightly points out differences in the size of different subsets of G.E. and the need to account for these in assessing enrichment of TAZ occupancy. Following the Reviewer's advice, we performed the re-shuffle separately for each of the G.E. subsets, namely i) all G.E., ii) G.E. in at least 5 PDOs, and iii) G.E. in at least 8 PDOs, in order to generate random sets that match in number and size the features in each of the G.E. subsets. Subsequently, we calculated the percentage of TAZ-bound regions in the random sets and compared them to the observed value of the original G.E. subsets. Notably, none of the 1000 reshuffled regions in each of the random sets had a percentage as high or higher than the observed percentage of TAZ-bound enhancer regions (**Figure R3**) confirming the results of the previous analysis. This revised analysis is now included as **Fig. 4g**.

Figure R3. Distribution of TAZ-bound reshuffled regions in the random sets generated for each of the G.E. subsets: i) all G.E., ii) G.E. conserved in at least 5 patients, and iii) G.E. conserved in at least 8 patients. The vertical lines denote the observed number of TAZ-bound enhancers for each of the above G.E. subsets.

Further analyses would also help strengthen the conclusion. The authors could, for instance, compare TAZ signal intensity at enhancers overlapping in 1/10, 5/10, 8/10 PDOs (heatmap and average signal) or TAZ binding in cancer vs normal tissue.

We thank the Reviewer for this suggestion. We compared the signal intensity of TAZ peaks that overlap conserved enhancers (shared by 8 or more PDOs) to that of TAZ peaks intersecting the remaining non-conserved G.E. (shared by < 5 PDOs). In agreement with the findings of the permutation analysis, we found that the levels of TAZ increase with the level of enhancer conservation across PDOs. This analysis is now included as **Supplementary Fig. 5c**.

2) Fig. 5: the results are very interesting, but to support the role of TAZ as a key factor it would be important to compare the set of TAZ-regulated enhancers to a control set where TAZ is not found (e.g. the GEs shared by 8/10 PDOs that do not overlap TAZ peaks, or GEs with a low average TAZ signal relative to input vs GEs with a higher average signal). This would allow to a) assess if TAZ binding is a defining feature of CRC and pan-cancer enhancers, b) evaluate the number of pan-cancer enhancers that can be identified independently of TAZ, and c) check if H3K27ac differences relative to normal tissues are e.g. larger for TAZ-regulated enhancers.

We thank the Reviewer for this interesting suggestion. We note that we do not wish to claim that only YAP/TAZ orchestrate the epigenetic alterations that occur in tumor cells, but rather that they are one of the major contributors. Although our work reveals that a large fraction of the conserved CRC enhancerome is regulated by the transcriptional coactivators YAP/TAZ, it is possible that additional players are involved in the epigenetic deregulation of the cancer cell state. We have now made this more explicit in the Discussion of our revised manuscript.

Following the reviewer's suggestion, we investigated the 291 gained enhancers in our dataset that are shared by 8/10 PDOs but not bound by TAZ in two ways: We first looked at H3K27ac levels in CRC PDOs relative to

normal tissues, and subsequently we evaluated the H3K27ac differences in 43 diverse primary tumors and normal tissues, as we did for the core of 46 TAZ-bound enhancers that are shared across diverse cancer types (**Fig. 5a-b**). We also considered the evaluation of pan-cancer enhancers. However, this analysis involved the integration of TAZ ChIP-seq data with TCGA ATAC-seq data based on the prior identification of TAZ as a key transcription factor. Thus, the implementation of this TF-focused strategy to the TAZ-independent enhancers is not immediate and will require extensive additional investigation which can be part of future work, beyond the scope of this manuscript.

Nevertheless, when we looked at H3K27ac signals, we found that the differences (fold change) were significantly greater ($P = 0.005$, Wilcoxon rank sum test) for TAZ-regulated than TAZ-independent enhancers in our PDOs versus normal tissues (**Figure R4**). Expanding our analyses to diverse primary tumors, we found a similar pattern of more pronounced differences relative to the normal samples in the TAZ-bound compared to the non-TAZ-bound enhancers (**Figure R5**). Overall, these new analyses on TAZ-independent enhancers support our original conclusion that TAZ is a major regulator of the epigenetic alterations that define the cancer cell state.

Figure R4. Boxplots showing the log₂ fold change of H3K27ac intensity in PDOs relative to normal tissues for TAZ-independent ($n = 291$; light brown) and TAZ-bound ($n = 195$; brown) gained enhancers. ** $P = 0.005$, Wilcoxon rank sum test.

Figure R5. Differences in the H3K27ac intensities of primary tumors relative to normal tissues from public available ChIP-seq data for (a) TAZ-bound enhancers (n = 195) and (b) TAZ-independent enhancers (n = 291). **** $P < 0.0001$, Wilcoxon rank sum test.

3) Fig. 6: in line with the previous comment, it would be informative to compute an expression signature including genes that are associated with TAZ-independent enhancers, and compare cell-to-cell variation for TAZ-independent genes to the CRB signature from TAZ-regulated pan-cancer genes. It would be similarly interesting to compare signatures from the pan-cancer set to genes associated with the CRC-specific set of TAZ-regulated enhancers, and assess differences among clones and subpopulations of cells.

We thank the Reviewer for this interesting suggestion. We evaluated the expression signature of genes annotated to the TAZ-independent enhancers (complementary CRB) in the scRNA-seq dataset of 23 CRC primary tumors. Consistent with the differential activation of the TAZ-independent enhancers in CRC PDOs versus normal colon tissues, the complementary CRB signature is also active in malignant cells compared to other cell populations. However, contrary to the TAZ-related signature the complementary CRB lacks specificity and shows high levels of activation also in non-malignant epithelial and stromal cells as shown in the tSNE graphs below (**Figure R6**). This bears testament to the central role of YAP/TAZ as regulators of a conserved and tumor-specific enhancerome.

Figure R6. t-SNE representation of the TAZ-independent score across 18,539 epithelial cells of tumor and normal colon tissues from 23 CRC patients. Contour lines denote malignant cells (purple), normal epithelial cells (green), and stromal cells (orange).

4) *Although not crucial, some parts of the methodology are explained very briefly and a more detailed description would improve the reproducibility of the results. For the differential expression analysis: are samples paired or unpaired? What group-level comparison and test was performed? For enhancer-promoter associations: what is the rationale for using different approaches? Is distance or overlap the criterion when Hi-C data is not used?*

We apologize that the methods were not sufficiently well described and we now clarified them in both the **Methods** and the **Results** sections. The differential expression analysis between PDOs and normal tissues was a paired analysis and we applied the default negative binomial method in DESeq. The enhancer-promoter associations were based primarily on chromosome conformation capture (Hi-C) data generated for human colon. For enhancers that were not annotated with capture Hi-C, we assigned a target gene by identifying the nearest protein-coding gene with an active ChromHMM-defined active promoter state. In the latter approach, we implemented the `annotatePeakInBatch` function (`output="both", PeakLocForDistance="middle"`) of `ChIPpeakAnno` (v3.12.7) identifying gene features with the nearest active TSS to the peak as well as gene features that overlap the peak. We updated the Methods section with a clearer description of the approaches used for annotating the enhancers.

REVIEWER COMMENTS

Reviewer #1 (Remarks to the Author):

Thanks to the additional analysis, the comments have been well addressed, and the conclusions strengthened.

Reviewer #2 (Remarks to the Author):

No more comments

Reviewer #3 (Remarks to the Author):

The revised manuscript does address my initial concerns and I recommend publication of this improved work.

Reviewer #4 (Remarks to the Author):

The authors have addressed all the points that I raised and I have no further comments.

Reviewer #5 (Remarks to the Author):

The authors addressed all my comments. In their response to other reviewers, the authors also included additional experiments that support their conclusions and corroborate previous results. Overall, I'm satisfied with the authors' reply and I would recommend the manuscript for publication.

Response to Reviewers' Comments

Epigenomic landscape of human colorectal cancer unveils an aberrant core of pan-cancer enhancers orchestrated by YAP/TAZ

We thank the Reviewers and Editors for their careful assessment of our work and insightful comments which helped us to improve the manuscript.

There were no remaining Reviewers' requests or comments to address.

Reviewer #1 (Remarks to the Author):

Thanks to the additional analysis, the comments have been well addressed, and the conclusions strengthen.

Reviewer #2 (Remarks to the Author):

No more comments

Reviewer #3 (Remarks to the Author):

The revised manuscript does address my initial concerns and I recommend publication of this improved work.

Reviewer #4 (Remarks to the Author):

The authors have addressed all the points that I raised and I have no further comments.

Reviewer #5 (Remarks to the Author):

The authors addressed all my comments. In their response to other reviewers, the authors also included additional experiments that support their conclusions and corroborate previous results. Overall, I'm satisfied with the authors' reply and I would recommend the manuscript for publication.